1         **A Hydrological Emulator for Global Applications – HE v1.0.0**

Yaling Liu[1,2], Mohamad Hejazi[1], Hongyi Li[3], Xuesong Zhang[1], Guoyong Leng[1]
[1]Joint Global Change Research Institute, Pacific Northwest National Laboratory, 5825
University Research Court, College Park, Maryland 20740, United States
[2] Department of Earth and Environmental Engineering, Columbia University, New York, NY
10027, United States
[3] Department of Land Resources and Environmental Sciences, Montana State University,
Bozeman, MT 59717, United States
Correspondence to: Yaling Liu (cauliuyaling@gmail.com)
**Abstract**
While global hydrological models (GHMs) are very useful in exploring water resources and
interactions between the Earth and human systems, their use often requires numerous model
inputs, complex model calibration, and high computation costs. To overcome these challenges,
we construct an efficient open-source and ready-to-use hydrologic emulator (HE) that can mimic
complex GHMs at a range of spatial scales (e.g., basin, region, globe). More specifically, we
construct both a lumped and a distributed scheme of the HE based on the monthly *"abcd"* model
to explore the tradeoff between computational cost and model fidelity.  Model predictability and
computational efficiency were evaluated in simulating global runoff from 1971-2010 with both
the lumped and distributed schemes. The results are compared against the runoff product from
the widely-used Variable Infiltration Capacity (VIC) model. Our evaluation indicates that the
lumped and distributed schemes present comparable results regarding annual total quantity,
spatial pattern and temporal variation of the major water fluxes (e.g., total runoff,
evapotranspiration) across the global 235 basins (e.g., correlation coefficient r between the
annual total runoff from either of these two schemes and the VIC is >0.96), except for several
cold (e.g., Arctic, Interior Tibet), dry (e.g., North Africa) and mountainous (e.g., Argentina)
regions. Compared against the monthly total runoff product from the VIC (aggregated from daily
runoff), the global mean Kling-Gupta efficiencies are 0.75 and 0.79 for the lumped and
distributed schemes, respectively, with the distributed scheme better capturing spatial
heterogeneity. Notably, the computation efficiency of the lumped scheme is two orders of
magnitude higher than the distributed one, and seven orders more efficient than the VIC model.
A case study of uncertainty analysis for the world's sixteen basins with top annual streamflow is
conducted using 100,000 model simulations, and it demonstrates the lumped scheme's
extraordinary advantage in computational efficiency. Our results suggest that the revised lumped
"*abcd*" model can serve as an efficient and acceptable HE for complex GHMs and is suitable for
broad practical use, and the distributed scheme is also an efficient alternative if spatial
heterogeneity is of more interest.

## 1 Introduction

A global hydrological model (GHM) is an effective tool to understand how water moves between soil, plants and the atmosphere. In terms of spatial discretization, hydrological models can be classified into: 1) lumped models treating one basin as a homogeneous whole and disregarding spatial variations, such as the Sacramento Soil Moisture Accounting Model (Burnash et al., 1973); and 2) distributed models where the entire basin is divided into small spatial units (e.g., square cells or triangulated irregular network) to capture spatial variability, such as the PCRaster Global Water Balance (Van Beek and Bierkens, 2009)  and the WASMOD-M (Widén-Nilsson et al., 2007). For simplicity, models with division of one basin into separate areas or sub-basins are also categorized as distributed ones here. The corresponding predictability and computational efficiency of GHMs may vary from model to model, due to difference in complexity and structure. Recent years have seen rapid progress in GHMs. They are widely used in assessing the impacts of climate change and land surface changes on the water cycle (Alcamo and Henrichs, 2002; Arnell and Gosling, 2013; Liu et al., 2013; Liu et al., 2014; Nijssen et al., 2001a), exploring spatial and temporal distribution of water resources (Abdulla et al., 1996; Alkama et al., 2010; Bierkens and Van Beek, 2009; Gerten et al., 2005; Tang et al., 2010), examining how human activities alter water demand and water resources (De Graaf et al., 2014; Döll et al., 2009; Hanasaki et al., 2008; Liu et al., 2015; Rost et al., 2008; Vörösmarty et al., 2000), and investigating the interactions between human activities and water availability by incorporating GHM with integrated assessment models (Kim et al., 2016).

Applying GHMs usually requires miscellaneous inputs, high computational costs, and a complex calibration process. These challenges stand out in practical situations, especially when the computational resources are limited. For instance, sensitivity analysis and uncertainty

quantification are often needed for decision making, but the users usually cannot afford to run a

large number of simulations with many GHMs like the VIC (also categorized as land surface

model (LSM)) due to their high computational expense (Oubeidillah et al., 2014). Another

situation is when the users seek reasonable estimates of water resources with minimal efforts

rather than acquiring highly accurate estimates through expensive inputs of time and efforts. For

example, when users seek to explore the hydroclimatology of a region and its long-term water

balance (Sankarasubramanian and Vogel, 2002), then GHMs with fine spatial (e.g., 1/8 degree)

and temporal resolution (e.g., hourly)  are not necessarily needed. In this case, simple models

that possess reasonable predictability and are computationally efficient tend to be more suitable.

In addition, some studies have shown that GHMs/LSMs are sometimes outperformed by simple

empirical statistical models (Abramowitz, 2005; Abramowitz et al., 2008; Best et al., 2015),

suggesting that some GHMs/LSMs may underutilize the information in their climate inputs and

that model complexity may undermine accurate prediction. This also indicates the potential

advantages of simple model over complex GHMs/LSMs. Thus, constructing simple models that

can emulate the dynamics of more complex and computational expensive models (e.g.,

GHMs/LSMs) is warranted.

The motivation of this work arises from the need to construct a hydrological emulator

(HE) that can efficiently mimic the complex GHMs to address the abovementioned issues for

practical use, which provides the opportunity of speeding up simulations at the cost of

introducing some simplification. We develop a HE that is easy-to-use and efficient for any

interested groups or individuals to assess water cycle at basin/regional/global scales. This HE

possesses the following features: 1) minimum number of parameters; 2) minimal climate input

that is easy to acquire; 3) simple model structure; 4) reasonable model fidelity that captures both

the spatial and temporal variability; 5) high computational efficiency; 6) applicable in a range of
spatial scales; and 7) open-source and well-documented.
To achieve our goal of identifying a suitable HE, we have explored many hydrological
models to find one that may meet our needs. We start with a simple baseline model characterized
by mean seasonal cycle; i.e., the inter-annual mean value for every calendar day (Schaefli &
Gupta, 2007). Among others, we also explore the *"abcd"* model because: 1) it is widely-used
and proven to have reasonable predictability (Fernandez et al., 2000; Martinez and Gupta, 2010;
Sankarasubramanian and Vogel, 2002; Sankarasubramanian and Vogel, 2003; Thomas, 1981;
Vandewiele and Xu, 1992; Vogel and Sankarasubramanian, 2003); 2) it uses a monthly time step
and requires less computational cost than daily or hourly models; 3) it has solid physical basis
hence has potential to be extended to other temporal scales (Wang and Tang, 2014); 4) it requires
minimal and easily-available inputs; 5) it only involves 4-7 parameters; and 6) it can simulate
variables of interest such as recharge, direct runoff and baseflow that many other simple models
can't simulate  (Vörösmarty et al., 1998) .  This study marks the first time that the "*abcd*" based
model is applied globally, and also the first time the predictability and computational efficiency
for both the lumped and distributed schemes are evaluated. Below we describe the baseline and
the "*abcd*" models and data in Section 2; and we present the evaluation of the two models,
discuss their appropriateness of serving as a HE in Section 3; finally, in Section 4 we summarize
this work with concluding remarks.

**2 Methods and data**
**2.1 Model description**
We examine two simple models – baseline and the "*abcd*" model (both lumped and
distributed scheme) in order to identify a suitable one for serving as a HE.
2.1.1 Baseline model
Following the work of Schaefli & Gupta (2007), we explore a baseline model
characterized by the inter-annual mean value for every calendar day, i.e., climatology. In this
study, the baseline model is based on monthly climatology runoff, which comes from a model
simulation product – i.e., the runoff product from the Variable Infiltration Capacity (VIC) model
(Leng et al. 2015). Specifically, we first calculate grid-level inter-annual mean value for each of
the 365 calendar days from daily runoff of the benchmark product during 1971-2010 (see Section
2.3.2), and then aggregate daily climatology runoff to monthly climatology runoff at grid-level.
The baseline model here uses monthly climatology runoff for prediction.  For example, if the
climatology runoff for July in one grid cell is 100 mm mon$^{-1}$, then the prediction of total runoff
for July of every year in that specific grid cell is 100 mm mon$^{-1}$.

2.1.2 The "*abcd*" model
The monthly *"abcd"* model was first introduced by Thomas (1981) to improve the national
water assessment for the U.S., with a simple analytical framework using only a few descriptive
parameters. It has been widely used across the world, especially for the U.S. (Martinez and
Gupta, 2010; Sankarasubramanian and Vogel, 2002; Sankarasubramanian and Vogel, 2003). The
model uses potential evapotranspiration (PET) and precipitation (P) as input. The model defines
four parameters $a$, $b$, $c$, and $d$ that reflect regime characteristics (Sankarasubramanian and Vogel,
2002; Thomas, 1981) to simulate water fluxes (e.g., evapotranspiration, runoff, groundwater
recharge) and pools (e.g., soil moisture, groundwater). The parameters $a$ and $b$ pertain to runoff
characteristics, and *c* and *d* relate to groundwater. Specifically, the parameter *a* reflects the
propensity of runoff to occur before the soil is fully saturated. The parameter b is an upper limit
on the sum of evapotranspiration (ET) and soil moisture storage. The parameter c indicates the
degree of recharge to groundwater and is related to the fraction of mean runoff that arises from
groundwater discharge. The parameter d is the release rate of groundwater to baseflow, and thus
the reciprocal of d is the groundwater residence time. Snow is not part of the original "*abcd*"
model, which may result in poor performance of the model in cold regions where snow
significantly affects the hydrological cycle. The work of Martinez and Gupta (2010) has added
snow processes into the original "*abcd*" model, where the snowpack accumulation and snow
melt are estimated based on air temperature.  Their work indicated that incorporation of the snow
processes in the monthly "*abcd*" model has significantly improved model performance in snow-
covered area in the conterminous United States (see Figure 4 in Martinez and Gupta (2010)).

In this study, we adopt the "*abcd*" framework from Martinez and Gupta (2010) (Fig. 1);

meanwhile, we make three modifications to suit the needs of a HE for global applications. First,
in order to enhance the model efficiency with as least necessary parameters as possible, instead
of involving three tunable snow-related parameters in the calibration process, we set the values
for two of the parameters (i.e., temperature threshold above or below which all precipitation falls
as rainfall or snow) from literature (Wen et al., 2013) and only keep one tunable parameter m –
snow melt coefficient ($0 < m < 1$). Second, we introduce the baseflow index (BFI) into the
calibration process to improve the partition of total runoff between the direct runoff and baseflow
(see Section 2.4). Third, other than the lumped scheme as previous studies used, we first explore
the values of model application in distributed scheme with a grid resolution of 0.5 degree. The
detailed model descriptions and equations are presented in the Appendix A, and the descriptions
and ranges of model parameters are listed in Table 1.

**2.2 Model structure**

In terms of the *"abcd"* model, we evaluate both the lumped and distributed model

schemes, although most previous applications of the model are conducted in a lumped scheme
(Bai et al., 2015; Fernandez et al., 2000; Martinez and Gupta, 2010; Sankarasubramanian and
Vogel, 2002; Sankarasubramanian and Vogel, 2003; Vandewiele and Xu, 1992; Vogel and
Sankarasubramanian, 2003). In the lumped scheme, each of the 235 river basins is lumped as a
single unit, and each of the data input (see Section 2.3.1) is the lumped average across the entire
basin, and thus all the model outputs are lumped as well. In terms of the distributed one, however,
each 0.5-degree grid cell has its own data inputs, and likewise, the model outputs are simulated
at the grid-level. Although the two schemes differ in the spatial resolution of their inputs and
outputs, their within-basin parameters are uniform. We use basin-uniform rather than grid-
specific parameters for the distributed scheme for two reasons: 1) to enhance computational
efficiency; and 2) to avoid drastically different parameters for neighboring grid cells that may be
unrealistic. Note that lateral flows between grid cells and basins are not included at this stage for
the *"abcd"* model. For the baseline model, as documented in Section 2.1.1, every 0.5-degree grid
cell of each basin has its own monthly climatology runoff estimates for each of the 12 calendar
months.

**2.3 Data**
2.3.1 Climate data
The climate data needed for the *"abcd"* model only involve monthly total precipitation,
monthly mean, maximum and minimum air temperature. The data we use is obtained from
WATCH (Weedon et al., 2011), spanning the period of 1971-2010, and it is 0.5-degree gridded
global monthly data. The climate data is used for model simulation over the global 235 major
river basins (Kim et al., 2016). Additionally, we use the Hargreaves-Samani method (Hargreaves
and Samani, 1982) to estimate potential evapotranspiration (PET), which is a required input for
the *"abcd"* model, and it needs climate data of mean, maximum and minimum temperatures for
the calculation.

2.3.2 Benchmark runoff product
In this study, the "*abcd*" model is tested for its ability to emulate the naturalized
hydrological processes of a reference model since the "true" naturalized hydrological processes
are unknown. The "perfect model" approach is well adopted in climate modeling studies where
one model is treated as "observations" while the others are tested for their ability to reproduce
"observations" (Murphy et al., 2004; Tebaldi and Knutti, 2007). Here, we use the process-based
VIC model as the "perfect model", which was also driven by the WATCH climate forcing.
The VIC runoff product here is a global simulation with a daily time step and spatial
resolution of 0.5 degree for the period of 1971-2010, and the VIC daily runoff is aggregated to
monthly data to be consistent with the temporal scale of the "*abcd*" model. The VIC model
settings used in this study are based on the University of Washington VIC Global applications
(http://www.hydro.washington.edu/Lettenmaier/Models/VIC/Datasets/Datasets.shtml). The sub-
grid variability of soil, vegetation and terrain characteristics are represented in sub-grid area-
specific parameter classifications. Soil texture and bulk densities are derived by combining the
World Inventory of Soil Emission Potentials database (Batjes, 1995) and the 5-min digital soil
map of the world from the Food and Agricultural Organization (FAO, 1998). Based on the work
of (Cosby et al., 1984), the remaining soil properties (e.g. porosity, saturated hydraulic
conductivity and unsaturated hydraulic conductivity) are derived. Vegetation type data are
obtained from the global land classification of (Hansen et al., 2000). Parameters including the
infiltration parameter, soil layer depths and those governing the baseflow function were
calibrated for major global river basins and transferred to the global domain as documented in
(Nijssen et al., 2001b), based on which Zhang et al. (2014) and Leng et al. (2015) conducted
additional calibrations in the China domain. In this study, the VIC model was forced by WATCH
climate forcing at the daily time step (Weedon et al., 2011), based on the calibrated parameters
from Nijssen et al. (2001b), Zhang et al., (2014) and Leng et al., (2015). The simulated runoff
used in this study has recently been validated globally within the framework of the Inter-Sectoral
Impact Model Intercomparison Project and shows reasonable performance compared to other
hydrological models (Hattermann et al., 2017; Krysanova and Hattermann, 2017).

The VIC runoff product (Hattermann et al., 2017; Leng et al., 2015) is then used as a

benchmark for calibrating and validating the "*abcd*" model due to two reasons. First, VIC runoff
has been evaluated across many regions of the globe and is proved to be reasonably well
(Abdulla et al., 1996; Hattermann et al., 2017; Maurer et al., 2001; Nijssen et al., 1997; Nijssen
et al., 2001b). Second, the simulated monthly runoff by the "*abcd*" model is more representative
of "natural conditions" because human activities (e.g., reservoir regulations and upstream water
withdrawals) are currently not represented in the model. Thus it tends to be more reasonable to
compare the simulated runoff against the VIC natural runoff product rather than comparing
against observed streamflow data from stream gauges (Dai et al., 2009; Wilkinson et al., 2014).
Despite potential bias in the VIC runoff product, using it as a benchmark here is to demonstrate
the capability of the HE developed in this work to mimic complex GHMs. Furthermore, the
application of the HE is not tied to the VIC model and should be able to emulate other GHMs.
The VIC runoff product compares well to other products (see Fig. S1, S2), including the
University of New Hampshire/Global Runoff Data Centre (UNH/GRDC) runoff product (Fekete
and Vorosmarty, 2011; Fekete et al., 2002) and the global streamflow product (Dai et al., 2009).
The scatterplot pattern of the VIC long-term annual runoff product vs. the GRDC product
(GRDC, 2017) matches well with that of the UNH/GRDC runoff vs. the GRDC product
(streamflow is transferred to the same unit as runoff via dividing by the basin area), which means
the behavior of the VIC runoff product is similar to that of the UNH/GRDC product. Further, the
correlation coefficient of the VIC and the UNH/GRDC long-term annual runoff is as high as 0.83
across the global 235 basins (Fig. S2). This suggests the reasonableness of VIC runoff product,
because the UNH/GRDC runoff is calibrated with the GRDC observations. At the same time, the
discrepancies between the VIC runoff products and the streamflow products (Fig. S2) may be
attributed to human activities, such as reservoir regulations and upstream water withdrawals,
which are not embedded in the runoff but reflected in the streamflow. This is because the VIC
model simulates runoff at natural conditions, and then a stand-alone routing model can be used to
route these flows downstream (Nijssen et al., 2001b). The routing model may account for human
activities such as water extractions and reservoir operations (Haddeland et al., 2014). However,
here we use the VIC runoff under natural conditions as the benchmark product, which explains
the discrepancies between the VIC runoff and observed streamflow products.
Uncertainties arising from the runoff process in the VIC model should be acknowledged.
Implementation of different runoff generation schemes (e.g. TOPMODEL) within the same
modeling framework is an alternative that can be adopted in the future to explore the uncertainty
range. A recent inter-model comparison study shows that the VIC model falls within the range of
large model ensembles (Hattermann et al. 2017). Notably, groundwater and its interaction with
river and land surface are not represented in the model. Thus, the model may not be able to fully
capture the hydrologic responses in areas where lateral flow and the three way streamflow-
aquifer-land interactions are important. Further, vegetation dynamics and water management that
may affect runoff are not considered in the model simulations. Nonetheless, the use of the HE
documented here is not tied to the VIC, and it could be used to emulate other GHMs of interest.

**2.4 Model calibration**
Typically, most applications of the "*abcd*" model utilize single-objective optimization for
total runoff (or streamflow) during the calibration process to minimize the difference between
measured and simulated streamflow (Bai et al., 2015; Martinez and Gupta, 2010;
Sankarasubramanian and Vogel, 2002). While this may lead to a good fit for simulated total
runoff, however, it may result in inappropriate partition of total runoff between direct runoff and
baseflow. To improve the accuracy of the simulated total runoff and the partition between direct
runoff and baseflow, we introduce the baseflow index (BFI) into the objective function.
Unlike the baseline model, the "*abcd*" model requires a calibration step for reasonable
parameterization so as to enable good prediction. As mentioned above, we incorporate BFI into
the objective function during the calibration process. On one side, we maximize Kling-Gupta
efficiency (KGE) (Gupta et al., 2009), which is used as a metric to measure the accuracy of the
simulated total runoff relative to the VIC benchmark runoff. The KGE is defined as the
difference of unity and the Euclidian distance (ED) from the ideal point, thus we maximize KGE
through minimizing the ED. The KGE and ED are calculated as follows (Gupta et al., 2009):
$\qquad KGE = 1 - ED$ (1)
$\qquad ED = \sqrt{(r-1)^2 + (\alpha-1)^2 + (\beta-1)^2}$ (2)
$\qquad r = \dfrac{Cov_{so}}{\sigma_s \cdot \sigma_s}$ (3)
$\qquad \alpha = \dfrac{\sigma_s}{\sigma_o}$ (4)
$\qquad \beta = \dfrac{\mu_s}{\mu_o}$ (5)
where $r$, $\alpha$, $\beta$, and $Cov_{so}$ are relative variability, bias, correlation coefficient, and covariance
between the simulated and observed values (here we treat the VIC runoff as the observed),
respectively; $\mu$ and $\sigma$ represent the mean and standard deviation (subscript "s" and "o" stand for
simulated and observed values). On the other side, we also nudge the simulated BFI towards the
benchmark BFI (here we treat the benchmark BFI as the observed) – the mean BFI of the four
products from (Beck et al., 2013). Then, the objective function is as follows:
$\qquad \min(ED + abs(BFI_{obs} - BFI_{sim}))$ (6)
where $min$ stands for minimizing the value in the parenthesis, $abs$ represents absolute value, ED
is the Euclidian distance between the simulated and observed total runoff (Gupta et al., 2009),
$BFI_{obs}$ and $BFI_{sim}$ are the observed and simulated BFI, respectively. Here we treat the benchmark
runoff from the VIC and BFI from Beck et al. (2013) as observed values. We then minimize the
objective function for parameter optimization by utilizing a Genetic Algorithm (GA) routine
(Deb et al., 2002). Note that for the distributed model scheme, we aggregate the grid-level total
runoff estimates to basin-level and then nudge it toward basin-level benchmark total runoff
during the calibration process.

**2.5 Model simulations**

To evaluate the predictability and efficiency of the baseline and the "*abcd*" model so as

to identify a suitable one to serve as a HE, we have conducted a series of simulations.
Specifically, for the baseline model, no simulations are needed as it uses inter-annual mean value
for each month – 12 monthly values – as prediction, so we just replicate the 12 monthly runoff
for 1971-2010 and for each of the global 235 basins, and then compare against the benchmark
runoff product. For the "*abcd*" model, two sets of model simulations across the global 235 basins
are conducted, with one set for calibration and the other one for validation, for both the lumped
and distributed model schemes. For the first set, we run the model for each basin for the period
of 1971-1990 to get basin-specific parameters by using the GA approach (see Section 2.4). For
the second set, using the parameters identified in the first set of simulation, we run the model for
the period of 1991-2010 to validate the model predictability and also evaluate the computational
efficiency. Model inputs and outputs in the distributed scheme are at a spatial resolution of 0.5-
degree, whereas those in the lumped scheme are all in lumped single unit for each basin. All
model simulations are conducted in a monthly time step. Note that broad users can run the
identified HE for global 235 basins, or for as many basins as they want for either scheme, as all
the related basin-specific input data and calibrated parameters for both schemes are open-source.

**3 Results and discussions**

**3.1 Comparison of performances between the baseline and the "*abcd*" model**

Generally, we find baseline model performs worse than the "*abcd*" model (Fig. 2). The
baseline model exhibits a lower global mean KGE value (0.61) than the lumped and distributed
schemes of the "*abcd*" model (0.75 and 0.79, respectively). In addition, our analysis indicates
that the incorporation of BFI into the objective function leads to a significant improvement in the
partition of total runoff between direct runoff and baseflow (Fig. 3, Fig. S4), without
compromising predictability for total runoff, i.e., the global mean KGE values for modeled total
runoff with or without the incorporation of BFI are almost the same (0.75 vs 0.76). Specifically,
for the case of involving both the total runoff and BFI in the objective function, the correlation
efficiencies (r) between the long-term annual benchmark and modeled direct runoff, and between
benchmark and modeled baseflow from the lumped scheme across global basins are both 0.98
(Fig. 3), which are much higher than those of 0.86 and 0.72 in the case of only involving the total
runoff in the objective function (Fig. S4). Given the superiority of the "*abcd*" model over the
baseline model, we focus in the following sections on evaluating the predictability and
computational efficiency of the "*abcd*" model and its potential to serve as a HE.

**3.2 Evaluation of model predictability**
In terms of total runoff, we find the lumped and distributed schemes are comparably
capable in simulating long-term mean annual quantity, temporal variations and spatial patterns
for the vast majority of river basins globally (Fig. 3-5, Fig. S3). Estimates of long-term mean
annual total runoff from both the lumped and distributed schemes match very well with that of
VIC total runoff across the 235 basins, with a correlation coefficient (r) of higher than 0.96, for
both the calibration and validation period (Fig. 3). Similarly, the basin-level estimates of long-
term mean annual direct runoff and baseflow also match well with those of the VIC across the
globe, for both schemes and both periods (Fig. 3). This suggests both schemes possess the
capability in partitioning total runoff.

Furthermore, both schemes display good capability in capturing the seasonal variations of

the total runoff for both the calibration and validation period (Fig. 4, Fig. S5). Meanwhile,
although the spatial patterns of annual total runoff from the lumped scheme present a general
match with that of the VIC, it does not reflect the spatial variations inside a basin that is however
captured by the distributed scheme (Fig. 5). Likewise, overall much lower percentage differences
between the modeled runoff from the distributed scheme and the VIC runoff product than those
between the VIC and the lumped one further corroborate the significantly better performance of
the distributed scheme (Fig. S6). Both schemes still show large percentage differences in some
dry (e.g., North Africa) or cold regions (e.g., Tibet Plateau). This is because the runoff there is at
a low magnitude and thus small changes in runoff will lead to large percentage differences.
Therefore, the distributed scheme provides overall slightly higher KGE (Fig. 6), with a global
mean KGE value of 0.79 as compared to 0.75 for the lumped scheme (Fig. 2).

To ensure good model predictability for the major water fluxes, we also evaluate the

modelled ET estimates. The modelled ET compares reasonably well with the VIC ET product as
well as with the mean synthesis of the LandFlux-EVAL ET product (Mueller et al., 2013),
displaying similar spatial variations (Fig. S7). Likewise, the distributed *"abcd"* scheme tends to
have better capability in presenting spatial heterogeneity than the lumped one. In addition, the
percentage differences between our modeled ET and the VIC ET product further confirm that the
distributed scheme significantly outperforms the lumped one (Fig. S8), with much lower
differences from the VIC ET product, although discrepancies still exist in some extremely cold
(e.g., Greenland) or dry regions (e.g., North Africa), which is because small differences in ET
will lead to large percentage difference in those regions with low ET. Further, given the changes
in basin-scale monthly soil moisture is relatively small, precipitation should approximate the sum
of ET and runoff according to the water mass balance, the good predictability of seasonality in
runoff as illustrated in Fig. 4 also reflects similar performance for ET.

The distributed scheme appears to outperform the lumped scheme in term of goodness-

of-fit, especially in some cold (e.g., Arctic, Northern European, Interior Tibet) and in some dry
(e.g., North Africa) regions (Fig. 6). This is possibly because distributed inputs can reflect basin-
level heterogeneity, and thus better capture the characteristic of the hydrological conditions in
those regions. However, both schemes do not perform well in the southern end of the Andes
Mountains (Fig. 6). This may be attributed to the complex land surface characteristics in that
mountainous area, which cannot be resolved due to the coarse spatial resolution. Moreover, the
distributed scheme seems not performing very well in some cold regions (Fig. 6), which is
possibly due to lack of representation for permafrost in the model.

Previous studies investigating the credibility of lumped and distributed hydrological

models indicate that, in many cases, lumped models perform comparably or just as well as
distributed models (Asadi, 2013; Brirhet and Benaabidate, 2016; Ghavidelfar et al., 2011;
Michaud and Sorooshian, 1994; Obled et al., 1994; Reed et al., 2004; Refsgaard and Knudsen,
1996; YAO et al., 1998). However, distributed models may have advantages for predicting
runoff in ungauged watersheds (Reed et al., 2004; Refsgaard and Knudsen, 1996), for capturing
spatial distribution of runoff due to heterogeneity in rainfall patterns or in land surface (Downer
et al., 2002; Paudel et al., 2011; YAO et al., 1998). Our results on the predictability of lumped
and distributed "*abcd*" model are in line with previous findings in the literature.
The good agreement between our modelled water fluxes, including total runoff, direct
runoff, baseflow and ET, and the benchmark products provides confidence in the capability of
both the lumped and distributed schemes in estimating temporal and spatial variations in major
water fluxes across the globe. In addition, to identify a suitable HE, the required computation
cost is another key factor as detailed below.

**3.3 Evaluation of computational efficiency**
While the performance of model predictability is comparable for the lumped and
distributed schemes as elucidated above, great disparities still exist for runtime of the two
schemes and the VIC model (Table S1). Take the Amazon basin that covers a total number of
2002 0.5-degree grid cells as an example, it takes 11.05 minutes for model calibration via the GA
method for the distributed scheme but only 0.16 minute for the lumped one. Similar disparity is
also found for model simulation with calibrated parameters, with runtime of 0.03 and 3.20
seconds for a 1000-year simulation of the Amazon basin for the lumped and distributed schemes,
respectively. However, according to the authors' experience, it will take ~1 week for the VIC
model to accomplish the same job, which is far more computationally expensive. In general, the
computational efficiency of the lumped scheme is two orders of magnitudes higher than the
distributed one, although that of the distributed one is still much higher than the VIC (~five
orders of magnitude) and many other GHMs and LSMs. Note that all of the simulations here are
conducted on the Pacific Northwest National Laboratory (PNNL)'s Institutional Computing (PIC)
Constance cluster using 1 core (Intel Xeon 2.3 GHz CPU) with the same configuration.

**3.4 Potential application of the "*abcd*" model as a hydrological emulator**
The good predictability and computational efficiency of both the distributed or lumped
schemes as elucidated in Sections 3.2 and 3.3 suggest its suitability for serving as HEs that can
efficiently emulate complex GHMs (e.g., the VIC or others). The source codes, input data, basin-
specific parameters across the globe for both the lumped and distributed schemes are open-
source and well-documented, which will make the HE ready to use and facilitate their wide and
easy use with minimal efforts.
The choice of either the distributed or lumped scheme as HE depends on the user's
specific needs. There is a tradeoff between the model predictability and computational efficiency.
While the distributed scheme tends to better capture the spatial heterogeneity of water fluxes and
can produce grid-level outputs that lumped scheme cannot, it incurs higher computational cost
than the lumped scheme. For applications that aim to strike a balance between predictability and
computation cost, such as practical assessment of water resources, or estimation of water supply
for integrated assessment models (IAMs), or quantification of uncertainty and sensitivity
analyses, it would be reasonable to employ the lumped scheme as a HE. The lumped scheme is
especially advantageous due to its minimal calibration and computational cost, parsimonious
efforts for model implementation, and reasonable fidelity in estimating major water fluxes (e.g.,
runoff, ET). For users from the IAM community, the lumped scheme might be sufficiently
suitable for their needs since 1) the lumped scheme can operate at the same spatial resolution at
which IAMs typically balance water demands and supplies (Edmonds et al., 1997; Kim et al.,
2006; Kim et al., 2016), and 2) the inherent uncertainty of the lumped scheme is likely
comparable or even overshadowed by the intrinsic uncertainty of IAMs (Kraucunas et al., 2015;
O'Neill et al., 2014). Similarly, for users who aim to conduct uncertainty and sensitivity analyses,
the high computational efficiency of the lumped scheme allow the users to emulate the
hydrological model of interest (e.g., GHMs, LSMs) and then run a large number of simulations
to conduct their uncertainty and sensitivity analysis (Scott et al., 2016). Therefore, the high
computational efficiency makes the lumped scheme more appealing as a HE in these cases.
However, if the research questions hinge on the gridded estimates, or emphasize the spatial
heterogeneity of the water fluxes or pools, it would be more desirable to deploy the distributed
scheme as a HE instead. For example, a follow-up work is coupling the distributed scheme of the
HE with a widely-used IAM, the Global Change Assessment Model (GCAM, Edmonds et al.,
1997), and then using the coupled model to investigate the impacts of a variety of land use
policies on global water scarcity, where the HE is used to estimate grid-level runoff globally
under different land use policies.

While many studies indicate that basin runoff generation is sensitive to factors such as

physical characteristics, spatiotemporal variability in storage distribution and forcing input,
evidence also show that basin response can be captured using a handful of parameters (Hsu et al.,
1995; Young and Parkinson, 2002). In this study, the lumped scheme of the HE ignores the
spatiotemporal variability in basin characteristics by averaging the input forcing data;
consequently, the associated responses in within-basin runoff or ET variations cannot be
captured. In contrast, the distributed scheme presents a better performance in capturing
spatiotemporal variability of runoff and ET with use of the same input data, and without
increasing the number of parameters. Thus, the use of the distributed scheme is preferred when
the tradeoff in the computational efficiency is not a constraining factor.

Moreover, a combination of a top-down approach (Sivapalan et al., 2003) and a multi-

objective approach to model evaluation (Gupta et al., 1998) could be used to explore internal
basin behavior, wherein the top-down approach would start from a simple structure and then
progressively expand based on its caveats in reproducing overall basin behavior [e.g.,
Jothityangkoon et al., 2001]. In this study we adopt a similar framework, by starting from a
baseline model and then expanding to the "abcd" model with snow representation, also by
incorporating the baseflow index into the objective function to exert a multi-objective approach.
Our assessment indicates that a baseline model characterized by mean seasonal cycle still holds a
promise in predicting runoff at basins with small variability in basin characteristics, such as
basins of Ob, Lena, Yenisey, Siberia and Mackenzie in the Arctic area, where the baseline model
yields KGE values of greater than 0.90 from our evaluation. Further, while Martinez and Gupta
(2010) indicated that the incorporation of the snow component and an additional snow parameter
into the original "abcd" model has greatly improved model performance in snow-prevailed
regions, areas without prevailing snow (e.g., tropical zone) could still utilize the original version
of the "abcd" model to keep the model as parsimonious as possible without compromising model
predictability. In addition, although our results reveal that incorporation of baseflow index into
the objective function generally improves the model performance in partitioning of runoff
between direct runoff and baseflow, simply employing a single-objective approach (i.e., only
involving total runoff) also works well for some basins such as North Interior Africa and Interior
Australia. Thus, the single-objective approach is also acceptable for those basins with the
advantage of simplicity without compromise in performance. In short, according to specific basin
characteristics and the research needs, suitable model complexity and number of parameters
could be identified by following abovementioned scenarios, such that either the baseline model
or a reduced format of the HE (e.g., without snow representation or single-objective) could be
potentially utilized with the merits of simplicity, reasonable predictability and computational
efficiency, rather than adopting the full format of the HE. Future research can extend this work
by systematically investigating the role of different levels of inputs, parameters, and model
complexity on model performance in different basins across the globe.

Based upon our open-source HE and the validated basin-specific parameters across the

globe, researchers can easily investigate the variations in water budgets at the basin/
regional/global scale of interest, with minimum requirements of input data, efficient computation
performance and reasonable model fidelity. Likewise, researchers can utilize the framework of
the HE with any alternative input data, or recalibrate the HE to emulate other complex GHMs or
LSMs of interest, to meet their own needs.

**3.5 Case study for uncertainty analysis**

To demonstrate the capability of the examined "*abcd*" model serving as a HE, we use the

lumped scheme to conduct parameter-induced uncertainty analysis for the runoff simulation at
the world's sixteen river basins with top annual flow (Dai et al. 2009). Specifically, for each of
the sixteen basins, we first apply ±10% change to each of the five calibrated parameters (a, b, c,
d, m) to compose varying ranges; note that we just truncate the range to those valid in Table 1 if
the ±10% change exceeds the valid range. Then we randomly sample the five parameters from
corresponding ranges for 100,000 times (i.e., 100,000 combinations of parameters). After that,
we run the lumped scheme 100,000 times for each basin with the 100,000 combinations of
parameters to examine the parameter-induced uncertainty in total runoff. The uncertainty
analysis indicates that most basins are robust to changes in parameters, other than the Tocantins,
Congo and La Plata (Fig. 7). In other words, for basins Congo and La Plata, slight changes in
parameters may lead to large changes in runoff estimates. Then the uncertainty in the calibrated
parameters for the two basins may lead to large bias in the simulated runoff, which may more or
less explain why modeled runoff for the two basins tend to have higher biases than other basins
(Fig. 4). Notably, the 100,000 times of simulations only takes ~80 seconds on a Dell Workstation
T5810 with one Intel Xeon 3.5 GHz CPU, which demonstrates the extraordinary computational
efficiency of the lumped scheme and its advantage for serving as a HE.

**4 Conclusions**
Toward addressing the issue that many global hydrological models (GHMs) are
computationally expensive and thus users cannot afford to conduct a large number of simulations
for various tasks, we firstly construct a hydrological emulator (HE) that possesses both
reasonable predictability and computation efficiency for global applications in this work. Built
upon the widely-used *"abcd"* model, we have adopted two snow-related parameters from
literature rather than tuning them for parameter parsimony, and also have improved the partition
of total runoff between the direct runoff and baseflow by introducing baseflow index into the
objective function of the parameter optimization. We then evaluate the appropriateness of the
model serving as an emulator for a complex GHM – the VIC, for both the lumped and distributed
model schemes, by examining their predictability and computational efficiency.
In general, both distributed and lumped schemes have comparably good capability in
simulating spatial and temporal variations of the water balance components (i.e., total runoff,
direct runoff, baseflow, evapotranspiration). Meanwhile, the distributed scheme has slightly
better performance than the lumped one (e.g., capturing spatial heterogeneity), with mean Kling-
Gupta efficiency of 0.79 vs. 0.75 across global 235 basins, and also it provides grid-level
estimates that the lumped one incapable of. Additionally, the distributed scheme performs better
in extreme climate regimes (e.g., Arctic, North Africa) and Europe. However, the distributed one
incurs two more orders of magnitudes of computation cost than the lumped one. A case study of
uncertainty analysis with $100,000$ simulations for each of the world's sixteen basins with top
annual streamflow further demonstrates the lumped scheme's extraordinary advantage in terms
of computational efficiency. Therefore, the lumped scheme could be an appropriate HE –
reasonable predictability and high computational efficiency. At the same time, the distributed
scheme could be a suitable alternative for research questions that hinge on grid-level spatial
heterogeneity. Finally, upon open-sourcing and well-documentation, the HE is ready to use and it
provides researchers an easy way to investigate the variations in water budgets at a variety of
spatial scales of interest (e.g., basin, region or globe), with minimum requirements of efforts,
reasonable model predictability and extraordinary computational efficiency.

**Code and/or data availability**

The hydrological emulator (HE) is freely available on the open-source software site

GitHub (https://github.com/JGCRI/hydro-emulator/). We have released the version of the

specific HE v1.0.0 referenced in this paper on https://github.com/JGCRI/hydro-

emulator/releases/tag/v1.0.0, where the source code (written in Matlab), all related inputs,

calibrated parameters and outputs for each of the global 235 basins, as well as the user's manual

are available. In addition, the HE documented here has been translated into Python and is being

incorporated into Xanthos (Li et al., 2017), which is an open-source global hydrologic model that

allows users to run different combinations of evapotranspiration, runoff, and routing models. The

HE will be the default runoff model used in Xanthos 2.0 and will be available on GitHub

(https://github.com/JGCRI/xanthos).

**Appendix A: Descriptions and equations of the "*abcd*" model**

The *abcd* model was first introduced by (Thomas, 1981), and Martinez and Gupta (Martinez and Gupta, 2010) added snow processes into the model. In this work, we adopted the snow scheme in Martinez and Gupta (2010):

$$
Snow_i =
\begin{cases}
0 & T^{rain} < T_i^{\min} \\[2ex]
P_i \times \dfrac{T^{rain} - T_i^{\min}}{T^{rain} - T^{snow}} & T^{snow} < T_i^{\min} < T^{rain} \\[2ex]
P_i & T_i^{\min} < T^{snow}
\end{cases}
\tag{1}
$$

$$
SP_i = SP_{i-1} - SNM_i + Snow_i
\tag{2}
$$

$$
SNM_i =
\begin{cases}
0 & T_i^{\min} < T^{snow} \\[2ex]
(SP_{i-1} + Snow_i) \times m \times \dfrac{T^{rain} - T_i^{\min}}{T^{rain} - T^{snow}} & T^{snow} < T_i^{\min} < T^{rain} \\[2ex]
(SP_{i-1} + Snow_i) \times m & T^{rain} < T_i^{\min}
\end{cases}
\tag{3}
$$

where $P_i$, $SP_i$, $SNM_i$ and $Snow_i$ are total precipitation, snowpack storage, snowmelt and the precipitation as snowfall at time step $i$, respectively, $T^{rain}$ (or $T^{snow}$) stands for the temperature threshold above (or below) which all precipitation falls as rainfall (or snow), and $T_i^{\min}$ is the minimum temperature at time step $i$, and the parameter $m$ is the snowmelt coefficient. Rather than keeping the three parameters $T^{rain}$, $T^{snow}$ and $m$, we adopt the $T^{rain}$ value of 2.5 °C and $T^{snow}$ value of 0.6 °C (Wen et al., 2013) and thus only keep one snowmelt-related parameter $m$

in the model, in order to alleviate the computation load during the parameter optimization
process.

The model defines two state variables "available water" and "evapotranspiration

opportunity", denoted as $W_i$ and $Y_i$, respectively. The $W_i$ is defined as:
$W_i = SM_{i-1} + Rain_i + SNM_i$                                   (4)
where $SM_{i-1}$ is soil moisture at the beginning of time step $i$, $Rain_i$ and $SNM_i$ are rainfall and
snowmelt during period $i$.

$Y_i$ stands for the maximum water that can leave the soil as evapotranspiration (*ET*) at

period $i$, and it is defined as below:
$Y_i = ET_i + SM_i$                                                  (5)
where $ET_i$ is the actual ET at time period $i$ and $SM_i$ is soil moisture at the end of time step $i$.
Further, $Y_i$ has a non-linear relationship with $W_i$ as:
$Y_i = \dfrac{W_i - b}{2a} - \sqrt{(\dfrac{W_i - b}{2a})^2 - W_i \times b / a}$                  (6)
where a and b are parameters detailed in Section 2.1.2.

Allocation of $W_i$ between $ET_i$ and $SM_i$ is estimated by assuming that the loss of soil

moisture by *ET* will be proportional to potential evapotranspiration (*PET*) as:
$\dfrac{dS}{dt} = -PET \times \dfrac{SM}{b}$                         (7)
where *PET* is calculated by using the Hargreaves-Samani method (Hargreaves and Samani,

1982).

After integrating the above differential equation and assuming $S_{i-1} = Y_i$, $SM_i$ can be derived as:
$SM_i = Y_i \times \exp(^{-PET_i}/_b)$                               (8)
Then, $ET_i$ can be calculated through equation (2).

In the model framework, $W_i - Y_i$ is the sum of the groundwater recharge ($RE$) and direct

runoff ($Q_d$), and the allocation is determined by the parameter c:
$RE_i = c \times (W_i - Y_i)$                                                                 (9)
$Q_d = (1-c) \times (W_i - Y_i)$                                                        (10)
The baseflow from the groundwater ($GW$) pool is modeled as:
$Q_b = d \times GW_i$                                                                 (11)
where d is a parameter reflecting the release rate of groundwater to baseflow. Then the total
runoff ($Q_t$) is the sum of the direct runoff and baseflow:
$Q_t = Q_d + Q_b$                                                                    (12)
The $GW_i$ is the sum of groundwater storage at the end of last time step and the groundwater
recharge minus the baseflow, and $GW_i$ is derived as:
$GW_i = \dfrac{GW_{i-1} + RE_i}{1 + d}$                                                      (13)
Then, all the water fluxes and pools are solved.
Author contribution
Yaling Liu and Mohamad Hejazi designed this work, and all co-authors offered help through
discussions. Yaling Liu developed the hydrological emulator and conducted the simulations and
evaluations. Yaling Liu wrote the manuscript, and all co-authors contributed to the revision.
**Competing interests**
The authors declare that they have no conflict of interests.
**Acknowledgement:** This research was supported by the Office of Science of the U.S.
Department of Energy through the Integrated Assessment Research Program. PNNL is operated
for DOE by Battelle Memorial Institute under contract DE-AC05-76RL01830. We thank Chris
Vernon for his help in maintaining the Github repository for the hydrological emulator
(https://github.com/JGCRI/hydro-emulator/).

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

 **Figure Caption**

 **Figure 1** Schematic diagram of the "*abcd*" model, with enhancements of snow and partition of

 total runoff between direct runoff and baseflow.

 **Figure 2** Kling-Gupta efficiency of the simulated basin-level total runoff across the global 235

 basins (lump = lumped, dist = distributed, cal = calibration, the x-axis labels of "lump_cal" or

 "dist_cal" represent lumped/distributed scheme during calibration period).

 **Figure 3** Comparison of basin-specific long-term annual total runoff, direct runoff and baseflow

 estimates from both the lumped and distributed "*abcd*" model schemes against VIC products,

 across global 235 basins and for the calibration period of 1971-1990 and validation period of

 1991-2010. The labels are denoted as combination of model scheme and period, where lump and

 dist stand for lumped and distributed model scheme, cal and val represent the calibration and

 validation period, respectively. These denotations remain the same for all figures in this work.

 Note that the basin-level VIC baseflow is derived by multiplying the gridded VIC long-term

 annual total runoff and the mean of the four gridded baseflow index products from Beck et al.

 (2014), and then aggregating from grid-level to basin-level. The basin-level VIC direct runoff is

 then calculated by subtracting baseflow from the total runoff.

 **Figure 4** Time series of basin-specific total runoff ($Q_{total}$) from the VIC product, the lumped and

 distributed "*abcd*" schemes for the world's sixteen river basins with top annual flow (Dai et al.

 2009) during 1981-1990 (part of the calibration period 1971-1990). $KGE_l$ and $KGE_d$ stand for

 KGE value for the lumped and distributed scheme, respectively.

 **Figure 5** Spatial patterns of long-term annual total runoff (mm $yr^{-1}$) during 1971-1990 across

 global 235 basins: a) VIC runoff product; b) total runoff estimates from the lumped "*abcd*"

scheme (lump = lumped); and c) total runoff estimates from the distributed "*abcd*" scheme (dist
= distributed).
**Figure 6** The spatial pattern of Kling-Gupta efficiency (KGE) for the total runoff estimates of
the global 235 basins for the calibration period of 1971-1990: a) the lumped "*abcd*" scheme; and
b) the distributed "*abcd*" scheme.
**Figure 7** Parameter-induced uncertainty in total runoff for the world's sixteen river basins with
top annual flow. The green line stands for simulated total runoff using the calibrated parameters,
and the gray area represents the spread derived from variations in parameters.
Figure 1

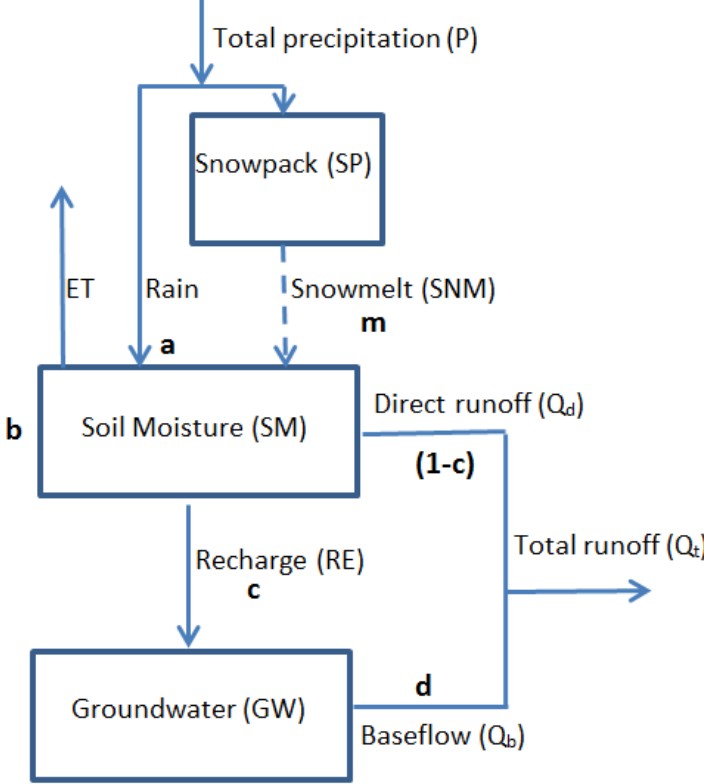



Figure 2

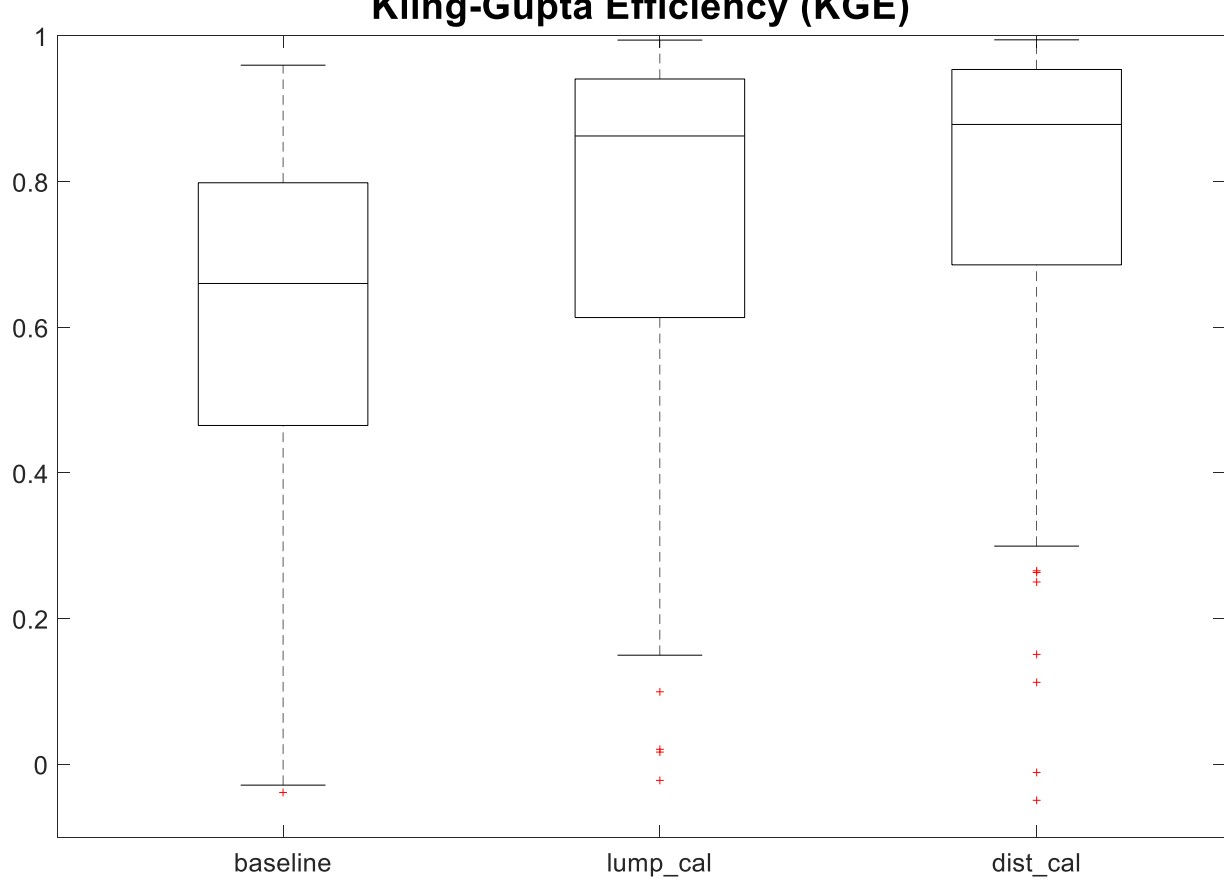


Figure 3

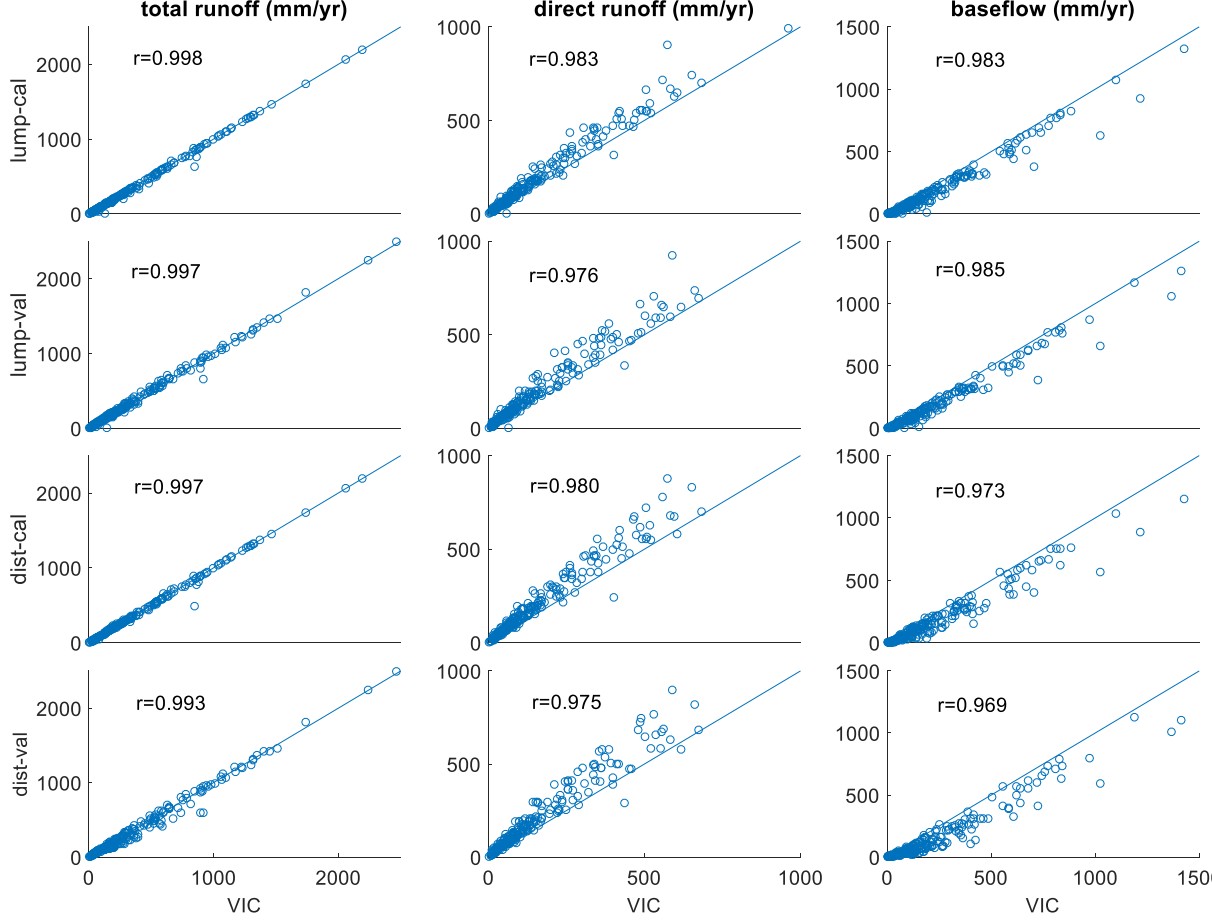



Figure 4

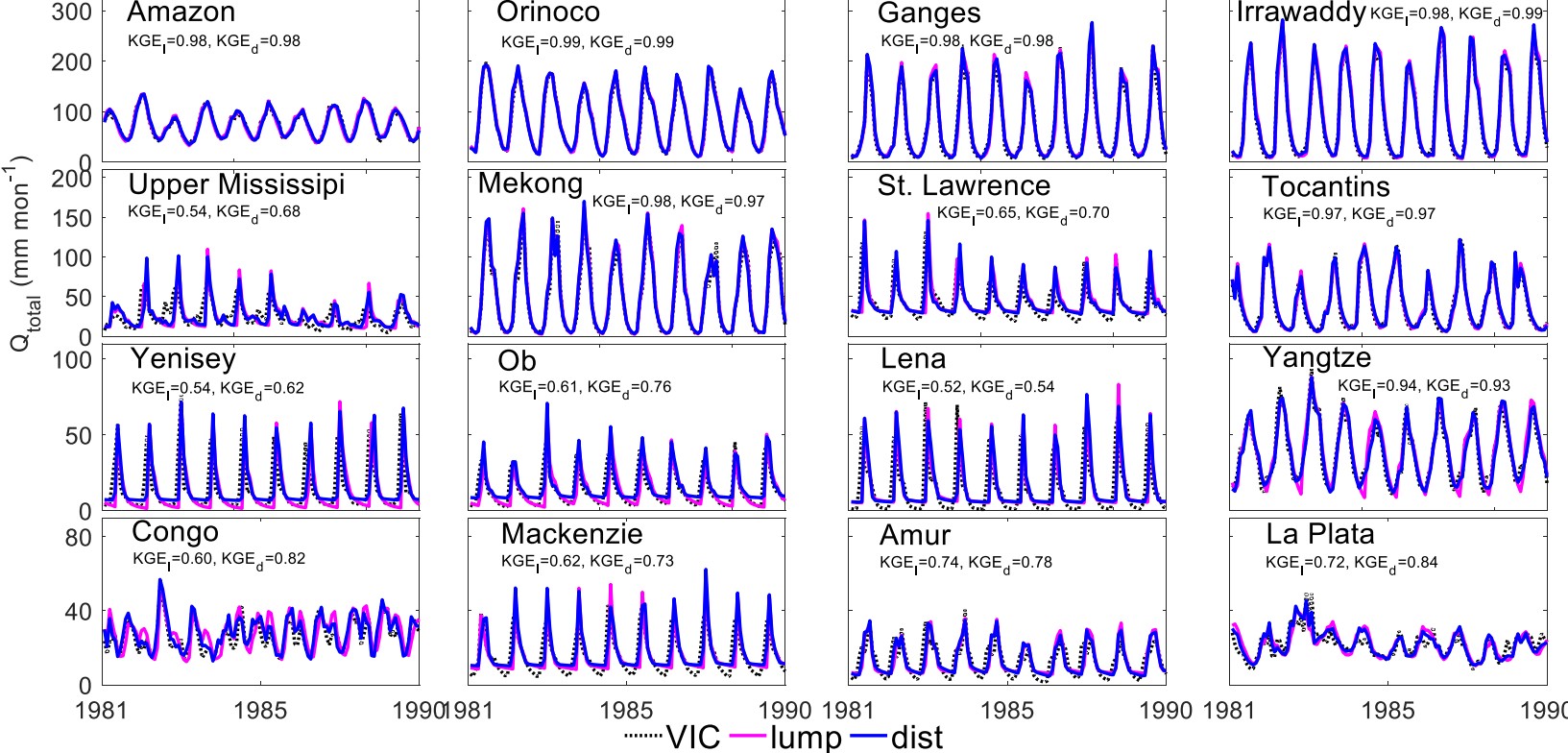

Figure 5

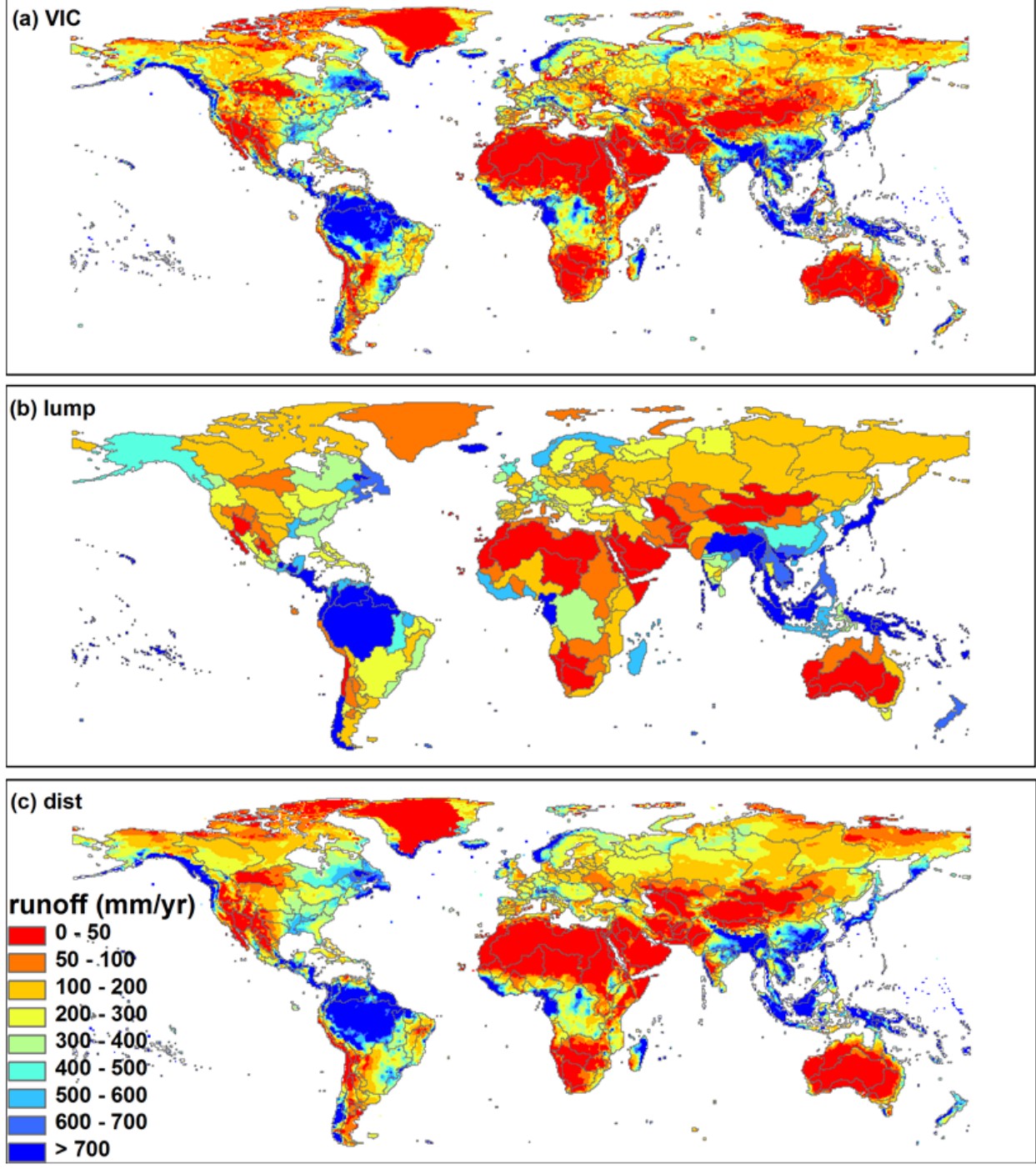

Figure 6

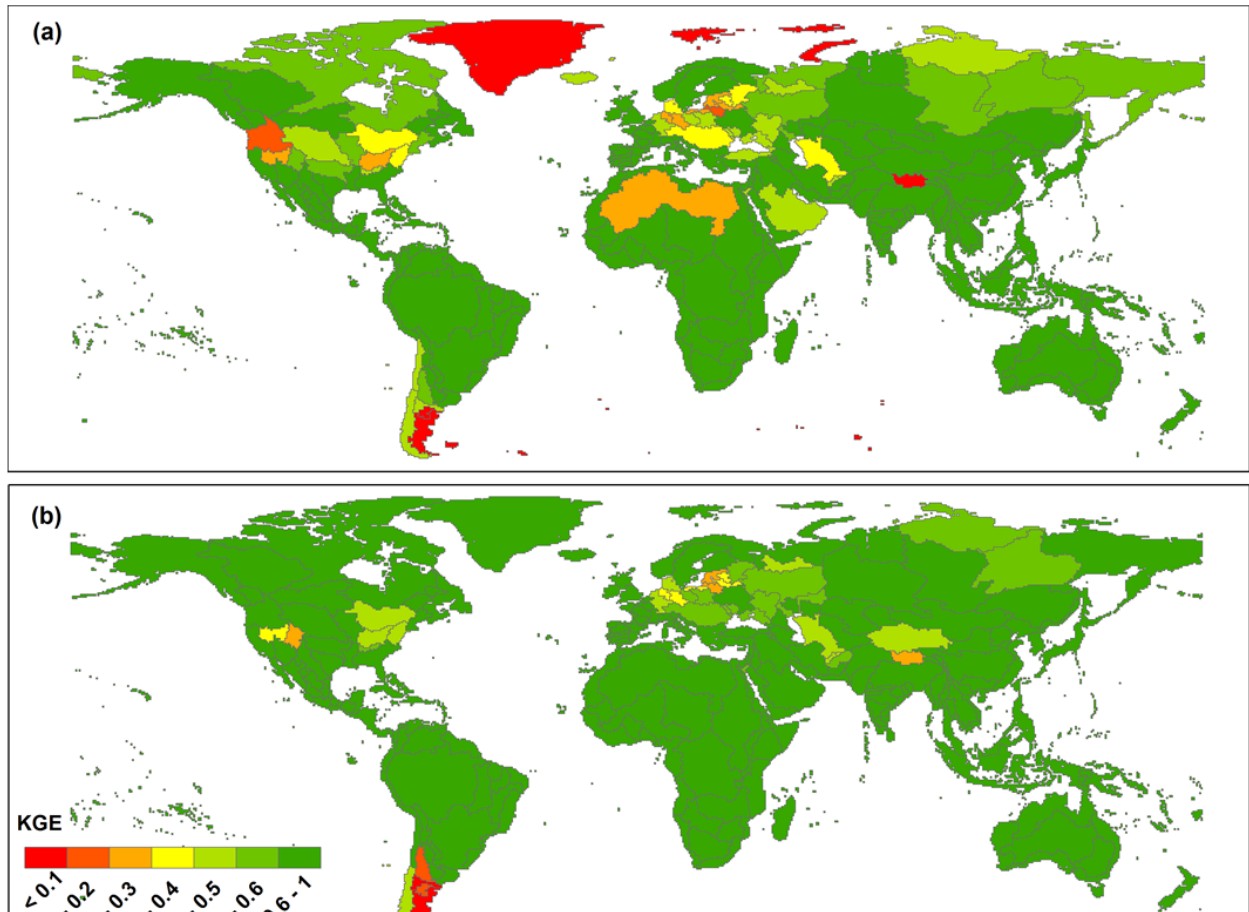

Figure 7

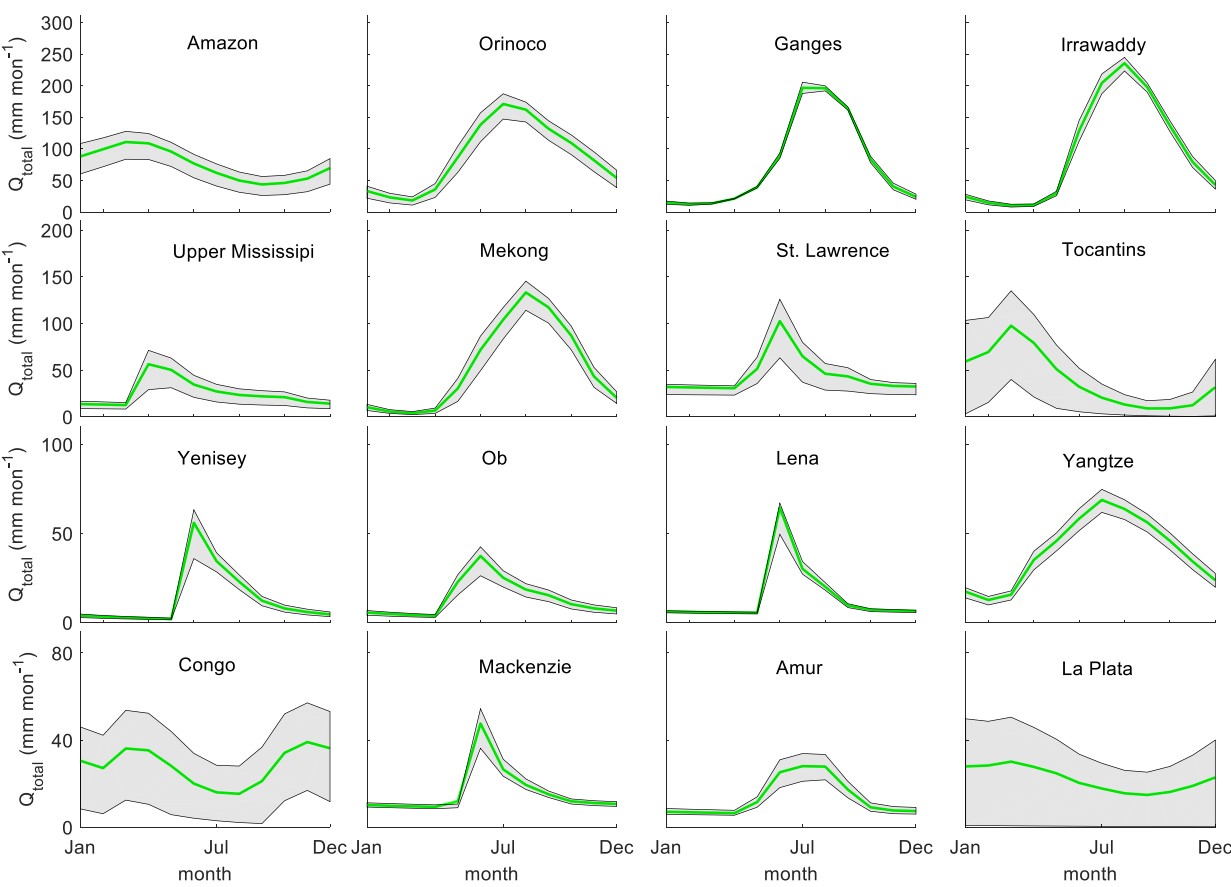

Table 1 Parameters description and ranges for the "*abcd*" model (the parameters *a,c,d* and *m* are dimensionless, and the unit for parameter *b* is mm)

| parameter | description | range | references |
|---|---|---|---|
| *a* | Propensity of runoff to occur before the soil is fully saturated | 0-1 | (Alley, 1984; Martinez and Gupta, 2010; Sankarasubramanian and Vogel, 2002; Vandewiele and Xu, 1992) |
| *b* | Upper limit on the sum of evapotranspiration and soil moisture storage | 0-4000 | |
| *c* | Degree of recharge to groundwater | 0-1 | |
| *d* | Release rate of groundwater to baseflow | 0-1 | |
| *m* | Snow melt coefficient | 0-1 | (Wen et al., 2013) |