# Peer review of "A Hydrological Emulator for Global Applications – HE v1.0.0"

_Geoscientific Model Development, 2017_

## Short Comment (SC1) · 14 Jul 2017

Dear Authors,

in my role as Executive editor of GMD, I would like to bring to your attention our Editorial version 1.1:

http://www.geosci-model-dev.net/8/3487/2015/gmd-8-3487-2015.html

This highlights some requirements of papers published in GMD, which is also available on the GMD website in the 'Manuscript Types' section:

http://www.geoscientific-model-development.net/submission/manuscript_types.html

From your abstract and introduction I understand that you describe the newly developed Hydrological Emulator and evaluate its results. Therefore your paper is not an

"Evaluation paper" but a "Development and Technical paper" and thus the criteria of this paper type are applied.

These are in particular:

- "The main paper must give the model name and version number (or other unique identifier) in the title."

- "All papers must include a section, at the end of the paper, entitled 'Code availability'. Here, either instructions for obtaining the code, or the reasons why the code is not available should be clearly stated. It is preferred for the code to be uploaded as a supplement or to be made available at a data repository with an associated DOI (digital object identifier) for the exact model version described in the paper. Alternatively, for established models, there may be an existing means of accessing the code through a particular system. In this case, there must exist a means of permanently accessing the precise model version described in the paper. In some cases, authors may prefer to put models on their own website, or to act as a point of contact for obtaining the code. Given the impermanence of websites and email addresses, this is not encouraged, and authors should consider improving the availability with a more permanent arrangement. After the paper is accepted the model archive should be updated to include a link to the GMD paper."

Therefore please provide a version number and preferably the acronym used within the article (HE). Additionally, we strongly recomment to make the exact code version, your article refers to, available via a permanent archive providing a DOI (e.g. Zenodo).

Best regards,

Astrid Kerkweg

---

## Referee Comment (RC1) · Anonymous Referee #1 · 17 Jul 2017

The manuscript by Liu et al. addresses the interesting issue of model complexity needed for global hydrological simulations. They present a new simulation tool based on the existing *abcd* model, and show that their simulations show a fair performance when compared with simulations from the VIC model. While I am generally supportive of work aimed at finding optimum model complexities, I feel the current study will need additional work to further show and quantify the benefits of the current code. At the moment, the main message seems to be that a low-dimensional model can produce positive correlations at the monthly timescale with another model, and that the runtime of the simple model is shorter. Both findings are not particularly new, and, in my view, they are not enough to merit publication. The suggested benefits of a simpler model (the possibility of focussing on uncertainty and spatial heterogeneity) might be true, but none of this is actually shown in the paper and no model or code is presented that takes full advantage of these suggested benefits. I believe the authors should present

more work in this direction before the manuscript can be accepted for publication in GMD.

My main concerns are the following:

- The motivation for choosing the *abcd* model is poor. Many simple models exist, and no objective criteria were used to select this particular model. The authors could have started with a simpler version, and adding components/complexity until a pre-defined threshold performance was reached. This would have made the selection less arbitrary. How does the modelled runoff for instance compare to a baseline "model" which is simply the monthly $P-PET$? The choice for the *abcd* model should be motivated better, but preferably a more systematic approach should be taken.

- The notion that simple models can do a good job in describing the output of more complex models is not new. In particular, Gab Abramovic has written numerous papers on this topic. This work should be considered and used in the interpretation/motivation.

- The motivation for the study is weak. In the current work, the authors only show a single application of their model (at grid and basin scales) and argue this is a good alternative to more complex models. But why not use the output of these complex models directly if the main goal is a best assessment of monthly average predictions of water balance partitioning? Such (multi-model) output is readily available at the global scale and does not require the running of even a simple model. Of course a simple model can be used for sophisticated uncertainty assessment (important advantage), but the authors did not yet do any work in this direction. This should be part of a revised version.

- The choice for the VIC model is poorly motivated. While I agree that some studies have shown that VIC produces positive NSE scores against observations,

many of these studies evaluated their results at very course time resolutions at which nearly any model would show a good performance (in particular because at monthly timescales the seasonal cycle dominates, which is easy to reproduce). The VIC model will generally not work well when evaluated at hourly or daily timesteps, even when calibrated. Related to this point is the issue of temporal resolution. It can be questioned whether nonlinear processes such as snow accumulation and melt can be modelled at a monthly timestep and at course spatial scales (see Melsen et al., *Hydrol. Earth Syst. Sci.* doi:10.5194/hess-20-1069-2016). In order to show that this is indeed possible, the authors should show that their model is able to outperform a baseline model consisting of, for instance, a mean seasonal cycle (as in Schaefli & Gupta, *Hydrol. Process.* 21, 2075–2080).

---

## Referee Comment (RC2) · T Roy (Referee) · 19 Jul 2017

In this study, the authors use a simple hydrological model "abcd" to emulate the behavior of more complex models (e.g. VIC). They modify the abcd model by including the baseflow index to better represent the partition of total runoff into direct runoff and baseflow. They present a lumped and a distributed version of the model, which are calibrated using the GA technique. They apply the model on global scale and compare the results against VIC simulations. Based on the results, they provide recommendations on the use of different versions of the model.

Although the model used is not new and the concept of simplified emulator is an established one, however, the global-scale application of the model and its assessment over multiple basins across globe make it an interesting study. A simple and computationally efficient emulator that can work well on global scale is useful for several applications.

[Figure]

I think the manuscript at its current stage needs some more work. Some additional analyses need to be added. Therefore, I suggest moderate revisions for the manuscript before it is accepted in GMD.

Following are my comments:

[1] How reliable are the VIC simulations? Calibration of VIC can significantly change its streamflow outputs. So what type of simulations are used in this case for the comparison purpose? How were the soil and vegetation parameters calibrated/selected? All these points need to be discussed in greater details.

[2] How did VIC perform in the extreme climate regions, for example, in snow-dominated catchments? This issue needs to be addressed properly. Maybe you can explore the following cases:

Case-1: If both emulator (E) and model (M) are matching the observations (O) well then that's great. There could be some sub-cases for this case:

(i) Both emulator and model match the observations well but from different directions (M – O – E). For example, they might have opposite (positive/negative) bias errors but the absolute values of the errors could be close.

(ii) The model is matching the observations well and the emulator is matching the model well, all in one direction (O – M – E).

(iii) The emulator is matching well both the model and the observations, but in different directions (M – E – O).

Case-2: If none of them are matching the observations well but their own outputs match each other, then too, I think an emulator is serving its purpose in a way (although not quite useful).

Case-3: If the emulator is matching the observations well but the model isn't then that's an interesting finding.

[Figure]

Case-4: If the model is matching the observations well but the emulator isn't then there is a problem. Therefore, this needs to be explored in greater depth.

[3] At seasonal time scales, the model performance is expected to be better. It would be crucial to also check the results on daily time scale. Maybe you can produce a set of time series plots, scatter plots, and spatial contour plots for daily level, as done for the seasonal case.

[4] Figure 3: Any idea why there are those biases in the lower streamflow values? Is there any location-specific pattern of these biases?

[5] Line 113: Which one's the other parameter you adopt the value of?

[6] Line 200: Did you try different weights on the two objectives?

[7] Line 290: In order to do a fair comparison, VIC and the two versions of the models should be run on the same computer, preferably with good configuration.

[8] Figure S1: I am not sure if you can say that all of them are comparing well. The discrepancies/mismatches should be clearly discussed in the manuscript. You are only showing the correlations here. What about the bias error?

[9] X-axis marks are missing for the first two subplots of Figure S1. Use same axis for the scatter plots in Figure 2.

[10] My comments about the manuscript:

Writing: The manuscript is very well written. I don't have any suggestions on this part.

Figures: Figures look good. Increase the legend in Figure 3.

Tables: Table 2 can go to the supplementary materials.
* * *

---

## Author Comment (AC1) · 19 Sep 2017

Response Letter

Title:       A Hydrological Emulator for Global Applications – HE v1.0.0

Journal:     Geoscientific Model Development

*We would like to thank the Editor and the referees for their detailed review of our manuscript and their positive feedback, constructive suggestions and criticisms. The responses to the Referees' comments are shown in* blue *font below. All the line numbers indicated refer to the main text of the revised manuscript (clean version without tracking changes).*

**Editor's comments:**
* * *
In my role as Executive editor of GMD, I would like to bring to your attention our Editorial version 1.1:

http://www.geosci-model-dev.net/8/3487/2015/gmd-8-3487-2015.html

This highlights some requirements of papers published in GMD, which is also available on the GMD website in the 'Manuscript Types' section:

http://www.geoscientific-model-development.net/submission/manuscript_types.html

From your abstract and introduction I understand that you describe the newly developed Hydrological Emulator and evaluate its results. Therefore your paper is not an "Evaluation paper" but a "Development and Technical paper" and thus the criteria of this paper type are applied.

These are in particular:

**Comment 1**: "The main paper must give the model name and version number (or other unique identifier) in the title."

Response 1: We thank the Editor for all the comments and for allowing us to revise the manuscript. We will change the "Manuscript type" to "Development and Technical paper" during our submission of the revision, and we have added model name and version number in the title:

"*A Hydrological Emulator for Global Applications – HE v1.0.0*"

**Comment 2:** "All papers must include a section, at the end of the paper, entitled 'Code availability'.

Here, either instructions for obtaining the code, or the reasons why the code is not available should be clearly stated. It is preferred for the code to be uploaded as a supplement or to be made available at a data repository with an associated DOI (digital object identifier) for the exact model version described in the paper. Alternatively, for established models, there may be an existing means of accessing the code

through a particular system. In this case, there must exist a means of permanently accessing the precise model version described in the paper. In some cases, authors may prefer to put models on their own website, or to act as a point of contact for obtaining the code. Given the impermanence of websites and email addresses, this is not encouraged, and authors should consider improving the availability with a more permanent arrangement. After the paper is accepted the model archive should be updated to include a link to the GMD paper."

Therefore please provide a version number and preferably the acronym used within the article (HE). Additionally, we strongly recomment to make the exact code version, your article refers to, available via a permanent archive providing a DOI (e.g. Zenodo).

Response 2: We have tried our best to meet the journal requirements in terms of code availability. First, we have created a repository in the open-source software site GitHub (https://github.com/JGCRI/hydro-emulator/) to make the hydrological emulator freely available. We have released the version of the specific HE v1.0.0 referenced in this paper on https://github.com/JGCRI/hydro-emulator/releases/tag/v1.0.0, where the source code (written in Matlab), all related data inputs and outputs, as well as the detailed Readme file are available. The repository is maintained by our organization, the Joint Global Change Research Institute (JGCRI), and long-term commitment for maintaining the repository is a standard practice. For example, both Le Page et al. (2016) and Hartin et al. (2015) published in Geoscientific Model Development (GMD) provided their codes on the GitHub site maintained by JGCRI (https://github.com/JGCRI/). Second, there is an ongoing effort to incorporate the hydrological emulator developed in this study to Xanthos (Li et al., 2017, https://github.com/JGCRI/xanthos), which is an open-source global hydrologic model, and the code for the HE referenced in this paper will also be freely available in the next version of Xanthos.

We have clarified it in the "Code and/or data availability" section as follows:

"*The hydrological emulator (HE) is freely available on the open-source software site GitHub (*https://github.com/JGCRI/hydro-emulator/). *We have released the version of the specific HE v1.0.0 referenced in this paper on* https://github.com/JGCRI/hydro-emulator/releases/tag/v1.0.0, *where the source code (written in Matlab), all related inputs, calibrated parameters and outputs for each of the global 235 basins, as well as the detailed Readme file are available.*"

References:

Hartin, C.A., Patel, P., Schwarber, A., Link, R.P. and Bond-Lamberty, B.P., 2015. A simple object-oriented and open-source model for scientific and policy analyses of the global climate system–Hector v1.0. Geoscientific Model Development, 8(4), pp.939-955.

Le Page, Y., West, T.O., Link, R., Patel, P., 2016. Downscaling land use and land cover from the Global Change Assessment Model for coupling with Earth system models. Geoscientific Model Development, 9(9), p.3055.

Li, X., Vernon, C.R., Hejazi, M.I., Link, R.P., Feng, L., Liu, Y., Rauchenstein, L.T., 2017, Xanthos – A Global Hydrologic Model, Journal of Open Research Software, 5(1), p.21.

---

## Author Comment (AC2) · 19 Sep 2017

**Response Letter to Reviewer #1**

Title:         A Hydrological Emulator for Global Applications – HE v1.0.0

Journal:      Geoscientific Model Development

*We would like to thank the referee for their detailed review of our manuscript and their positive feedback, constructive suggestions and criticisms. The responses to the Referee's comments are shown in* blue *font below. All the line numbers indicated refer to the main text of the revised manuscript (clean version without tracking changes).*

**Reviewers' Comments to Author:**

The manuscript by Liu et al. addresses the interesting issue of model complexity needed for global hydrological simulations. They present a new simulation tool based on the existing abcd model, and show that their simulations show a fair performance when compared with simulations from the VIC model. While I am generally supportive of work aimed at finding optimum model complexities, I feel the current study will need additional work to further show and quantify the benefits of the current code. At the moment, the main message seems to be that a low-dimensional model can produce positive correlations at the monthly timescale with another model, and that the runtime of the simple model is shorter. Both findings are not particularly new, and, in my view, they are not enough to merit publication. The suggested benefits of a simpler model (the possibility of focussing on uncertainty and spatial heterogeneity) might be true, but none of this is actually shown in the paper and no model or code is presented that takes full advantage of these suggested benefits. I believe the authors should present more work in this direction before the manuscript can be accepted for publication in GMD. My main concerns are the following:

**Comment 1**: The motivation for choosing the abcd model is poor. Many simple models exist, and no objective criteria were used to select this particular model. The authors could have started with a simpler version, and adding components/complexity until a pre-defined threshold performance was reached. This would have made the selection less arbitrary. How does the modelled runoff for instance compare to a baseline "model" which is simply the monthly P−PET? The choice for the abcd model should be motivated better, but preferably a more systematic approach should be taken.

Response 1: We have clarified the motivation for choosing the "*abcd*" model in lines 85-96:

"*To achieve our goal of identifying a suitable HE, we have explored many hydrological models to find one that may meet our needs. We start with a simple baseline model characterized by mean seasonal cycle; i.e., the inter-annual mean value for every calendar day (Schaefli & Gupta, 2007). Among others, we also explore the "abcd" model because: 1) it is widely-used and proven to have reasonable predictability (Fernandez et al., 2000; Martinez and Gupta, 2010; Sankarasubramanian and Vogel, 2002; Sankarasubramanian and Vogel, 2003; Thomas, 1981; Vandewiele and Xu, 1992; Vogel and Sankarasubramanian, 2003); 2) it uses a monthly time step and requires less computational cost than daily or hourly models; 3) it has solid physical basis hence has potential to be extended to other temporal scales (Wang and Tang, 2014); 4) it requires minimal and easily-available inputs; 5) it only involves 4-7 parameters; and 6) it can simulate variables of interest such as recharge, direct runoff and baseflow that many other simple models can't simulate  (Vörösmarty et al., 1998).*"

Further, in lines 137-146, we have described the modifications we have made to the "abcd" model:

"*In this study, we then adopt the "abcd" framework from Martinez and Gupta (2010) (Fig. 1); meanwhile, we make three modifications to suit the needs of a HE for global applications. First, in order to enhance the model efficiency with as least necessary parameters as possible, instead of involving three tunable snow-related parameters in the calibration process, we set the values for two of the parameters (i.e., temperature threshold above or below which all precipitation falls as rainfall or snow) from literature (Wen et al., 2013) and only keep one tunable parameter m – snow melt coefficient (0 < m < 1). Second, we introduce the baseflow index (BFI) into the calibration process to improve the partition of total runoff between the direct runoff and baseflow (see Section 2.4). Third, other than the lumped scheme as previous studies used, we first explore the values of model application in distributed scheme with a grid resolution of 0.5 degree.*"

We have also enhanced our analysis according to the referee's suggestions in terms of a simpler model in Comment 3 and 6. Specifically, we have added a baseline model to better justify the appropriateness of constructing the hydrological emulator (HE) based on the "*abcd*" model. We have added the descriptions for the baseline model in lines 107-114:

"*2.1.1 Baseline model*
        *Following the work of Schaefli & Gupta (2007), we explore a baseline model characterized by the inter-annual mean value for every calendar day, i.e., climatology. In this study, we adapt the baseline model to monthly scale by first calculating inter-annual mean value for every calendar day from daily runoff of the benchmark product during 1971-2010 (see Section 2.3.2), and then aggregating daily runoff to monthly runoff. The model uses climatology for prediction, for example, if the inter-annual mean runoff for July in the Amazon basin is 100 mm mon$^{-1}$, then the prediction of total runoff for July of every year is 100 mm mon$^{-1}$.*"

We have also added the comparison of performances between the baseline and the "abcd" model in lines 287-301 to elaborate its superiority over the baseline model:

"*Generally, we find baseline model performs worse than the "abcd" model (Fig. 2). The baseline model exhibits a lower global mean KGE value (0.61) than the lumped and distributed schemes of the "abcd" model (0.75 and 0.79, respectively). In addition, our analysis indicates that the incorporation of BFI into the objective function leads to significant improvement in the partition of total runoff between direct runoff and baseflow (Fig. S4), without compromising predictability for total runoff, i.e., the global mean KGE values for modeled total runoff with or without the incorporation of BFI are almost the same (0.75 vs 0.76). Specifically, for the case of involving both the total runoff and BFI in the objective function, the correlation efficiencies (r) between the long-term annual benchmark and modeled direct runoff, and between benchmark and baseflow from the lumped scheme across global basins are 0.97 and 0.96, respectively, which are much higher than those of 0.86 and 0.72 in the case of only involving the total runoff in the objective function (Fig. S4).*
        *Given the superiority of the "abcd" model over the baseline model, we focus in the following sections on evaluating the predictability and computational efficiency of the "abcd" model and its potential to serve as a HE.*"

[Figure]

Figure 2. Kling-Gupta efficiency of the simulated basin-level total runoff across the global 235 basins (lump = lumped, dist = distributed, cal = calibration, the x-axis labels of "lump_cal" or "dist_cal" represent lumped/distributed scheme during calibration period).

References:
Schaefli, B. and Gupta, H.V., 2007. Do Nash values have value? Hydrological Processes, 21(15), 2075-2080.
Fernandez, W., Vogel, R., Sankarasubramanian, A., 2000. Regional calibration of a watershed model. Hydrol. Sci. J., 45(5): 689-707.

Martinez, G.F., Gupta, H.V., 2010. Toward improved identification of hydrological models: A diagnostic evaluation of the "abcd" monthly water balance model for the conterminous United States. Water Resour. Res., 46(8).

Sankarasubramanian, A., Vogel, R.M., 2002. Annual hydroclimatology of the United States. Water Resour. Res., 38(6).

Sankarasubramanian, A., Vogel, R.M., 2003. Hydroclimatology of the continental United States. Geophys. Res. Lett., 30(7).

Thomas, H., 1981. Improved methods for national water assessment. Report WR15249270, US Water Resource Council, Washington, DC.

Vandewiele, G., Xu, C.-Y., 1992. Methodology and comparative study of monthly water balance models in Belgium, China and Burma. J. Hydrol., 134(1-4): 315-347.

Vogel, R.M., Sankarasubramanian, A., 2003. Validation of a watershed model without calibration. Water Resour. Res., 39(10).

Vörösmarty, C.J., Federer, C.A., Schloss, A.L., 1998. Potential evaporation functions compared on US watersheds: Possible implications for global-scale water balance and terrestrial ecosystem modeling. J. Hydrol., 207(3-4): 147-169.

Wang, D. and Y. Tang (2014), A one-parameter Budyko model for water balance captures emergent behavior in Darwinian hydrologic models, Geophysical Research Letters , 41, doi:10.1002/2014GL060509.

Wen, L., Nagabhatla, N., Lü, S., Wang, S.-Y., 2013. Impact of rain snow threshold temperature on snow depth simulation in land surface and regional atmospheric models. Adv. Atmos. Sci., 30(5): 1449-1460.

**Comment 2:** The notion that simple models can do a good job in describing the output of more complex models is not new. In particular, Gab Abramovic has written numerous papers on this topic. This work should be considered and used in the interpretation/motivation.

Response 2: We thank the referee for pointing out the useful references. We have added relevant papers of Gab Abramovic in the Introduction section (lines 69-75) to better justify our work of exploring a simple model:

"*In addition, some studies have shown that GHMs/LSMs are sometimes outperformed by simple empirical statistical models (Abramowitz, 2005; Abramowitz et al., 2008; Best et al., 2015), suggesting that some GHMs/LSMs may underutilize the information in their climate inputs and that model complexity may undermine accurate prediction. This also indicates the potential advantages of simple model over complex GHMs/LSMs. Thus, constructing simple models that can emulate the dynamics of more complex and computational expensive models (e.g., GHMs/LSMs) is warranted.*"

References:
Abramowitz, G., 2005. Towards a benchmark for land surface models. Geophys. Res. Lett., 32(22).

Abramowitz, G., Leuning, R., Clark, M., Pitman, A., 2008. Evaluating the performance of land surface models. J. Clim., 21(21): 5468-5481.

Best, M.J. et al., 2015. The plumbing of land surface models: benchmarking model performance. J. Hydrometeorol., 16(3): 1425-1442.

**Comment 3:** The motivation for the study is weak. In the current work, the authors only show a single application of their model (at grid and basin scales) and argue this is a good alternative to more complex models. But why not use the output of these complex models directly if the main goal is a best assessment

of monthly average predictions of water balance partitioning? Such (multi-model) output is readily available at the global scale and does not require the running of even a simple model. Of course a simple model can be used for sophisticated uncertainty assessment (important advantage), but the authors did not yet do any work in this direction. This should be part of a revised version.

Response 3: The main merit of a hydrological emulator (HE) is its capability of emulating complex global hydrological models (GHMs). Multi-model projects, such as ISI-MIP, do provide outputs like global runoff, but the available products are still very limited. The HE developed in this study provide an easy-to-use and open-source tool for the community to emulate GHMs of interest and simulate any scenarios of interest with reasonable predictability and high computational efficiency, which is a capability that is computationally prohibitive for multi-model projects using GHMs. Some related explanation has been added in lines 215-218:

"*Despite potential bias in the VIC runoff product, using it as a benchmark here is to demonstrate the capability of the HE developed in this work to mimic complex GHMs. Furthermore, the application of the HE is not tied to the VIC model and should be able to emulate other GHMs.*"

We also have clarified the usage of the HE in lines 399-404:
"*Based upon our open-source HE and the validated basin-specific parameters across the globe, researchers can easily investigate the variations in water budgets at the basin/ regional/global scale of interest, with minimum requirements of input data, efficient computation performance and reasonable model fidelity. Likewise, researchers can utilize the framework of the HE with any alternative input data, or recalibrate the HE to emulate other complex GHMs or LSMs of interest, to meet their own needs.*"

Further, we have followed the reviewer's suggestion and have conducted an uncertainty analysis (UA) to demonstrate the advantage of the HE. We have added the UA in Section 3.5 as follows:

"*3.5 Case study for uncertainty analysis*
*To demonstrate the capability of the examined "abcd" model serving as a HE, we use the lumped scheme to conduct parameter-induced uncertainty analysis for the runoff simulation at the world's sixteen river basins with top annual flow (Dai et al. 2009). Specifically, for each of the sixteen basins, we first apply ±10% change to each of the five calibrated parameters (a, b, c, d, m) to compose varying ranges; note that we just truncate the range to those valid in Table 1 if the ±10% change exceeds the valid range. Then we randomly sample the five parameters from corresponding ranges for 100,000 times (i.e., 100,000 combinations of parameters). After that, we run the lumped scheme 100,000 times for each basin with the 100,000 combinations of parameters to examine the parameter-induced uncertainty in total runoff. The uncertainty analysis indicates that most basins are robust to changes in parameters, other than the Tocantins, Congo and La Plata (Fig. 7). In other words, for basins Congo and La Plata, slight changes in parameters may lead to large changes in runoff estimates. Then the uncertainty in the calibrated parameters for the two basins may lead to large bias in the simulated runoff, which may more or less explain why modelled runoff for the two basins tend to have higher biases than other basins (Fig. 4). Notably, the 100,000 times of simulations only takes ~80 seconds on a Dell Workstation T5810 with one Intel Xeon 3.5 GHz CPU, which demonstrates the extraordinary computational efficiency of the lumped scheme and its advantage for serving as a HE.*"

[Figure]

Figure 7 Parameter-induced uncertainty in total runoff for the world's sixteen river basins with top annual flow. The green line stands for simulated total runoff using the calibrated parameters, and the gray area represents the spread derived from variations in parameters.

References:

Dai, A., Qian, T., Trenberth, K.E., Milliman, J.D., 2009. Changes in continental freshwater discharge from 1948 to 2004. J. Clim., 22(10): 2773-2792.

**Comment 4:** The choice for the VIC model is poorly motivated. While I agree that some studies have shown that VIC produces positive NSE scores against observations, many of these studies evaluated their results at very course time resolutions at which nearly any model would show a good performance (in particular because at monthly timescales the seasonal cycle dominates, which is easy to reproduce). The VIC model will generally not work well when evaluated at hourly or daily timesteps, even when calibrated. Related to this point is the issue of temporal resolution. It can be questioned whether nonlinear processes such as snow accumulation and melt can be modelled at a monthly timestep and at course spatial scales (see Melsen et al., Hydrol. Earth Syst. Sci. doi:10.5194/hess-20-1069- 2016). In order to show that this is indeed possible, the authors should show that their model is able to outperform a baseline model consisting of, for instance, a mean seasonal cycle (as in Schaefli & Gupta, Hydrol. Process. 21, 2075–2080).

Response 4: We agree with the reviewer about the performance of the VIC at fine time-steps (e.g., hourly). The essential point of this work is not to emulate VIC, but using VIC as an example to demonstrate the HE developed in this study could be used to emulate any global hydrological models (GHMs) of interest. We have clarified this in lines 215-218:

"*Despite potential bias in the VIC runoff product, using it as a benchmark here is to demonstrate the capability of the HE developed in this work to mimic complex GHMs. Furthermore, the application of the HE is not tied to the VIC model and should be able to emulate other GHMs.*"

Due to the requirement of high computational efficiency in addition to reasonable predictability, daily or sub-daily time step is not suitable for the HE, so we use monthly time step. In terms of the processes such as snow accumulation and melt, Martinez and Gupta (2010) have shown that the incorporation of snow processes in the monthly "*abcd*" model significantly improves the model performance in regions with snow cover. This is why we adopt the "*abcd*" version with the snow module (Martinez and Gupta 2010) in this study. We have clarified this in lines 132-136:

"*The work of Martinez and Gupta (2010) has added snow processes into the original "abcd" model, where the snowpack accumulation and snow melt are estimated based on air temperature. Their work indicated that incorporation of the snow processes in the monthly "abcd" model has significantly improved model performance in snow-covered area in the conterminous United States (see Figure 4 in Martinez and Gupta (2010)).*"

Other than that, we have followed the reviewer's suggestions and have added a baseline model in this work to reveal the superiority of the adopted "*abcd*" model over the baseline model, for details please see the Response 1.

References:

Martinez, G.F., Gupta, H.V., 2010. Toward improved identification of hydrological models: A diagnostic evaluation of the "abcd" monthly water balance model for the conterminous United States. Water Resour. Res., 46(8).

---

## Author Comment (AC3) · 19 Sep 2017

**Response Letter to Reviewer #2**

Title: A Hydrological Emulator for Global Applications – HE v1.0.0

Journal: Geoscientific Model Development

*We would like to thank the referee for their detailed review of our manuscript and their positive feedback, constructive suggestions and criticisms. The responses to the Referee's comments are shown in blue font below. All the line numbers indicated refer to the main text of the revised manuscript (clean version without tracking changes).*

**Reviewers' Comments to Author:**

In this study, the authors use a simple hydrological model "abcd" to emulate the behavior of more complex models (e.g. VIC). They modify the abcd model by including the baseflow index to better represent the partition of total runoff into direct runoff and baseflow. They present a lumped and a distributed version of the model, which are calibrated using the GA technique. They apply the model on global scale and compare the results against VIC simulations. Based on the results, they provide recommendations on the use of different versions of the model. Although the model used is not new and the concept of simplified emulator is an established one, however, the global-scale application of the model and its assessment over multiple basins across globe make it an interesting study. A simple and computationally efficient emulator that can work well on global scale is useful for several applications.

I think the manuscript at its current stage needs some more work. Some additional analyses need to be added. Therefore, I suggest moderate revisions for the manuscript before it is accepted in GMD. Following are my comments:

**Comment 1:** How reliable are the VIC simulations? Calibration of VIC can significantly change its streamflow outputs. So what type of simulations are used in this case for the comparison purpose? How were the soil and vegetation parameters calibrated/selected? All these points need to be discussed in greater details.

Response 1: The VIC runoff product (Hattermann et al., 2017; Leng et al. 2015) is used as a benchmark in this study, and its use is merely to demonstrate the capability of the hydrological emulator (HE) developed in this work to mimic complex global hydrological models (GHMs). Despite the potential bias in the VIC product, it does not affect the key findings of this work about the capability of the HE. We have added detailed descriptions about the VIC simulations in lines 186-206:

[revised manuscript text omitted]

Nijssen, B.N., G.M. O'Donnell, D.P. Lettenmaier and E.F. Wood, 2001: Predicting the discharge of global rivers, Journal of Climate, 14(15), 3307-3323

Weedon, G. et al., 2011. Creation of the WATCH forcing data and its use to assess global and regional reference crop evaporation over land during the twentieth century. J. Hydrometeorol., 12(5): 823-848.

Zhang, X. (2014). A long-term land surface hydrologic fluxes and states dataset for China. Journal of Hydrometeorology, 15(5), 2067-2084.

**Comment 2:** How did VIC perform in the extreme climate regions, for example, in snow dominated catchments? This issue needs to be addressed properly. Maybe you can explore the following cases: Case-1: If both emulator (E) and model (M) are matching the observations (O) well then that's great. There could be some sub-cases for this case: (i) Both emulator and model match the observations well but from different directions (M – O – E). For example, they might have opposite (positive/negative) bias errors but the absolute values of the errors could be close. (ii) The model is matching the observations well and the emulator is matching the model well, all in one direction (O – M – E). (iii) The emulator is matching well both the model and the observations, but in different directions (M – E – O). Case-2: If none of them are matching the observations well but their own outputs match each other, then too, I think an emulator is serving its purpose in a way (although not quite useful). Case-3: If the emulator is matching the observations well but the model isn't then that's an interesting finding. Case-4: If the model is matching the observations well but the emulator isn't then there is a problem. Therefore, this needs to be explored in greater depth.

Response 2: We thank the referee for the detailed comments regarding the comparison between the VIC model and the hydrological emulator (HE). The essential point of this work is to deliver an open-source and easy-to-use hydrological emulator that can be used for emulating global hydrological models (GHMs) of interest. VIC is used as an example GHM in this study to demonstrate the capability of the HE to emulate complex and computationally expensive GHMs (see also Response 4 in the Response letter to Reviewer #1). Exploring the sources of differences between the performance of the VIC and the HE is outside the focus of this work, and it would be incorporated in our future work.

**Comment 3:** At seasonal time scales, the model performance is expected to be better. It would be crucial to also check the results on daily time scale. Maybe you can produce a set of time series plots, scatter plots, and spatial contour plots for daily level, as done for the seasonal case.

Response 3: We agree with the reviewer on the better performance of monthly time scale than that of daily, however, due to the needs of high computational efficiency for the hydrological emulator (HE), it is simulated at monthly time step and a daily time series comparison is not even feasible in this case.

**Comment 4:** Figure 3: Any idea why there are those biases in the lower streamflow values? Is there any location-specific pattern of these biases?

Response 4: From the uncertainty analysis we added in Section 3.5 (see also Figure 7), it shows basins like Congo and La Plata are not as robust as other basins to changes in parameters – slight changes in parameters may lead to large changes in runoff estimates. Then the uncertainty in the calibrated parameters for the two basins may lead to large bias in the simulated runoff. We have added discussions in lines 415-421:

"*The uncertainty analysis indicates that most basins are robust to changes in parameters, other than the Tocantins, Congo and La Plata (Fig. 7). In other words, for basins Congo and La Plata, slight changes in parameters may lead to large changes in runoff estimates. Then the uncertainty in the calibrated parameters for the two basins may lead to large bias in the simulated runoff, which may more or less explain why modelled runoff for the two basins tend to have higher biases than other basins (Fig. 4).*"

**Comment 5:** Line 113: Which one's the other parameter you adopt the value of?

Response 5: We adapt two snow-related parameters from literature, and make the other one – snow melt coefficient – tunable during the calibration process, it has been clarified in lines 139-143:

"*in order to enhance the model efficiency with as least necessary parameters as possible, instead of involving three tunable snow-related parameters in the calibration process, we set the values for two of the parameters (i.e., temperature threshold above or below which all precipitation falls as rainfall or snow) from literature (Wen et al., 2013) and only keep one tunable parameter m – snow melt coefficient (0 < m < 1).*"

"*Note that all of the simulations here are conducted on the Pacific Northwest National Laboratory (PNNL)'s Institutional Computing (PIC) Constance cluster using 1 core (Intel Xeon 2.3 GHz CPU) with the same configuration.*"

**Comment 8:** Figure S1: I am not sure if you can say that all of them are comparing well. The discrepancies/mismatches should be clearly discussed in the manuscript. You are only showing the correlations here. What about the bias error?

Response 8: We thank the referee for the concern. Figure S1 is to illustrate the relationship of VIC and UNH/GRDC runoff product with streamflow measurements at gauge stations, and the similar scatter patterns between the upper and lower panel indicates the similarity of the two runoff products. This analysis is to reveal the appropriateness of the VIC runoff product as a benchmark product in this work. The discrepancies between runoff products and streamflow measurements are induced from the ignorance of river routing, reservoir regulations and upstream water withdrawals in the simulated runoff products. This has been recognized in the main text (lines 226-229):

"*At the same time, the discrepancies between the VIC runoff products and the streamflow products (Fig. S2) may be attributed to human activities, such as reservoir regulations and upstream water withdrawals, which are not embedded in the runoff but reflected in the streamflow.*"

However, exploring the sources and magnitudes of the discrepancies among them is outside the focus of this study.

**Comment 9:** X-axis marks are missing for the first two subplots of Figure S1. Use same axis for the scatter plots in Figure 2.

Response 9: The X-axis marks for Figure S1 have been fixed. For the previous Figure 2 (currently Figure 3 in the revised manuscript), we use different axis for total runoff, direct runoff and baseflow is because they have different magnitude, and this may make the figure and scatter points more discernable.

**Comment 10:** My comments about the manuscript: Writing: The manuscript is very well written. I don't have any suggestions on this part. Figures: Figures look good. Increase the legend in Figure 3. Tables: Table 2 can go to the supplementary materials.

Response 10: We have increased the legend in the previous Figure 3 (currently Figure 4 in the revised manuscript) and moved Table 2 to the supplementary materials as suggested.

---

## Referee Report (RR1)

Review of A Hydrological Emulator for Global Applications – HE v1.0.0, gmd-2017-113-manuscript-version2

This paper presents a hydrologic emulator (HE), built upon the pre-existing "abcd" model, designed for use in global modeling applications such as IAMs. Both a distributed, gridded version and a lumped, water basin scale version are described and evaluated. The HE is tested against a baseline model of climatological monthly mean runoff. The HE is calibrated and validated against the VIC model, and its computational efficiency is assessed. The development of a computationally efficient, open-source global hydrologic model emulator is timely and useful to the modeling community, as many inter-disciplinary multi-modeling studies are utilizing global hydrologic models. While this paper is well-written and will add a valuable model to the hydrology literature, there are some improvements that should be made before publication. These are described below, in addition to some suggestions.

Criticism related to previous reviewer's comments:

The current manuscript has successfully addressed most of the concerns of the previous reviews, but some improvements are still needed, and some concerns still need to be addressed.

1. The comparison of VIC runoff to GRDC data (Fig. S1) addresses the concern raised by a previous reviewer that VIC may not be an accurate model of global runoff. However, there needs to be a few improvements to this assessment:
- The acronyms GRDC and UNH/GRDC need their full names spelled out, and the GRDC needs to be properly cited.
- The top three panels of Fig. S1 look as though they are comparing UNH-GRDC (y-axis) to GRDC (x-axis). Such a comparison is not needed, and irrelevant, as the model UNH-GRDC product is calibrated to the GRDC data. Likely the figure is supposed to show VIC runoff vs GRDC runoff. Either the axis labels must be corrected, or the comparison needs to be redone.
- Why are these three basins chosen? The authors do not provide sufficient evidence that these basins are representative of global runoff patterns. The authors should either make this argument, or provide analysis of more basins. A map showing the $r^2$ values of monthly runoff in VIC vs GRDC or UNH-GRDC would be most informative, as it would show regions in which VIC is most (and least) accurate.
- Why is the comparison only made for the period 1986-1995? GRDC data now has observations through the year 2016.
- Suggestion: the authors could include a brief discussion of the limitations of the VIC model. This is not necessary, but could be helpful to readers.

2. The authors should assess the computational efficiency of the calibration processes. This would inform other users of the HE how difficult it is to re-calibrate

the model to other GHMs. While not necessary for publication, it would improve the paper to re-calibrate the HE to another GHM, demonstrating the HE's flexibility and broad applicability.

3. There is no analysis of daily runoff simulations.  Even if the model is not intended to be used for daily simulations, this should be explained explicitly in the text.  Line 342 states that distributed models such as the distributed HE presented here are better than lumped models for flood peak prediction.  However, flood peak prediction is only accurate at daily time steps, so this statement should either be removed, or the daily accuracy of the distributed HE assessed.

4. For context, the authors could add a brief description of the type of work that IAMs coupled with (or including) GHMs have been used for.

Major criticisms: must be addressed before publication

1. This model is intended to be fully open-source and user-friendly.  To accomplish this goal, the authors should include in the source package a user manual.  A good example of such a model user manual is the open source CaMa-Flood manual, available here: http://hydro.iis.u-tokyo.ac.jp/~yamadai/cama-flood/Manual_CaMa-Flood_v362.pdf
2. Line 107: Where does the baseline model's climatology runoff come from?  Is this based on data, or a model simulation?  It needs to be described and cited.
3. Lines 226 – 229: Provide data or a citation to back up the claim that discrepancies between VIC runoff and observed streamflow products are due to human activities.
4. If the VIC simulation did not include human activities, then can the HE model be used to emulate GHMs that do include human activities such as water extractions from rivers and reservoir operation?
5. Section 2.4: Please describe the runoff range over which the model is calibrated.  Does it include a good representation of extreme events?  How does the distribution of runoff in the calibration period compare to potential future runoff under climate change?  If there is a significant difference in these distributions, the applicability of the HE to climate change studies should be discussed.
6. Figure S4: Only the correlation coefficient for calibration on runoff is shown.  The correlation for calibration with runoff and BFI should be included, as it is discussed in the text.
7. Lines 318-325, and Fig. S5: While Fig. S5 shows maps of ET, there is no quantitative assessment of ET.  I suggest either a correlation analysis, or showing a difference map along with the other maps.  A difference map would be very informative, showing regions of good agreement and regions of poor agreement.
8. Lines 322 – 325, and Fig.4: Figure 4 shows a good match in seasonal variation of the calibration period. It is more important to show the seasonal

variation in the validation period.  The text claims that the seasonal variation in ET is good, but there is no quantitative evidence of this.

Minor criticisms: suggestions that are not essential for publication
1. Figure 4: The color scheme is good, as blue and black are similar, and the light green is hard to see.  Choosing different colors, or even using some dashed lines or other symbols would improve this figure.
2. Figure 5: Showing a difference map, especially between VIC and the distributed model, would be very informative.
3. While the citation for the PET calculation is given in the text, it would be useful to either cite this again within Appendix A, and/or provide the full equation for PET within Appendix A.
4. Line 257: The objective function equation needs an equation number.
5. While the paper is mostly well-written, the authors should have a copy editor review the paper for detailed grammatical issues, as there are several sprinkled throughout the text.  In a few places, these grammatical issues hinder the clarity of the text and should be revised. These places are:
    a. Lines 164 – 166, sentence beginning with "For the baseline model…"
    b. Lines 211 – 215, sentence beginning with "Second, since we have not…"
6. Lastly, the open source code is written in Matlab, a proprietary and costly computing software package.  While most large U.S. and European universities have Matlab licenses, this platform may be cost prohibitive to some researchers, limiting the global usability of the open source model.  While this is not required for publication, I would highly recommend that the authors translate this model into a fully open-source coding language such as R, Python, or C.

---

## Author Response (AR2)

Response Letter

Title:         A Hydrological Emulator for Global Applications – HE v1.0.0

Journal:    Geoscientific Model Development

*We would like to thank the Editor and the referees for their detailed review of our manuscript and their positive feedback, constructive suggestions and criticisms. The responses to the Referees' comments are shown in* blue *font below. All the line numbers indicated refer to the main text of the revised manuscript (clean version without tracking changes).*

**Editor's comments:**
* * *
**Topical Editor Decision**: Publish subject to minor revisions (review by editor) by Bethanna Jackson

Comments to the Author:

Thank you for the careful and attentive revision, and I second the comments from reviewers that the content is interesting and informative and a substantive contribution. Could you please add a small discussion on how catchment characteristics might change appropriate model structure/number of parameters as per Reviewer 1's suggestion, and give guidance on potential future work along this line, and also address the suggestions of Reviewer 2.

**Response 1**: We thank the Editor for allowing us to revise the manuscript. We have addressed the two Reviewers' comments accordingly (see responses below).

**Referee #1**

I enjoyed reading this revised manuscript. I felt that the authors have carefully addressed the comments from the two reviewers in the first round. The study has unique contribution in its global-scale examination of the HE and its comparison with a comprehensive land surface model (VIC). The HE included several new modifications from the "abcd" model, such as introducing a baseflow index to improve the partition of river flow into surface runoff and baseflow, and model implementation in both lumped and distributed schemes. Some issues still remain, such as selection of models and more detailed investigations for the differences in model performances, but I feel these could fall into another topic and can be addressed in different studies. I have one minor comment for the authors to consider before the paper is accepted for publication. In the

discussion section adding more information on what are the lessons learned from this excise? The watersheds at global scale differ so much in many aspects (e.g., snow vs. no snow, wet vs. dry, etc.). For some watersheds the model might be able to be further simplified without significantly sacrificing its performance, while for some other watersheds, adding more parameters/model components might be necessary to insure acceptable model performance. What are the recommendations? More information on this would help future model studies and the outcome of this study.

**Response 2**: We have added discussions on the application of the HE under different basin characteristics as follows (lines 432-469):

"*While many studies indicate that basin runoff generation is sensitive to factors such as physical characteristics, spatiotemporal variability in storage distribution and forcing input, evidence also show that basin response can be captured using a handful of parameters (Hsu et al., 1995; Young and Parkinson, 2002). In this study, the lumped scheme of the HE ignores the spatiotemporal variability in basin characteristics by averaging the input forcing data; consequently, the associated responses in within-basin runoff or ET variations cannot be captured. In contrast, the distributed scheme presents a better performance in capturing spatiotemporal variability of runoff and ET with use of the same input data, and without increasing the number of parameters. Thus, the use of the distributed scheme is preferred when the tradeoff in the computational efficiency is not a constraining factor.*

*Moreover, a combination of a top-down approach (Sivapalan et al., 2003) and a multi-objective approach to model evaluation (Gupta et al., 1998) could be used to explore internal basin behavior, wherein the top-down approach would start from a simple structure and then progressively expand based on its caveats in reproducing overall basin behavior [e.g., Jothityangkoon et al., 2001]. In this study we adopt a similar framework, by starting from a baseline model and then expanding to the "abcd" model with snow representation, also by incorporating the baseflow index into the objective function to exert a multi-objective approach. Our assessment indicates that a baseline model characterized by mean seasonal cycle still holds a promise in predicting runoff at basins with small variability in basin characteristics, such as basins of Ob, Lena, Yenisey, Siberia and Mackenzie in the Arctic area, where the baseline model yields KGE values of greater than 0.90 from our evaluation. Further, while Martinez and Gupta (2010) indicated that the incorporation of the snow component and an additional snow parameter into the original "abcd" model has greatly improved model performance in snow-prevailed regions, areas without prevailing snow (e.g., tropical zone) could still utilize the original version of the "abcd" model to keep the model as parsimonious as possible without compromising model predictability. In addition, although our results reveal that incorporation of baseflow index into the objective function generally improves the model performance in partitioning of runoff between direct runoff and baseflow, simply employing a single-objective approach (i.e., only involving total runoff) also works well for some basins such as North*

*Interior Africa and Interior Australia. Thus, the single-objective approach is also acceptable for those basins with the advantage of simplicity without compromise in performance. In short, according to specific basin characteristics and the research needs, suitable model complexity and number of parameters could be identified by following abovementioned scenarios, such that either the baseline model or a reduced format of the HE (e.g., without snow representation or single-objective) could be potentially utilized with the merits of simplicity, reasonable predictability and computational efficiency, rather than adopting the full format of the HE.* Future research can extend this work by *systematically investigating the role of different levels of inputs, parameters, and model complexity on model performance in different basins across the globe.*"

References:

Gupta, H. V., S. Sorooshian, and P. O. Yapo (1998), Toward improved calibration of hydrologic models: Multiple and noncommensurable mea- sures of information, Water Resour. Res., 34, 751–763.

Hsu, K., H. V. Gupta, and S. Sorooshian (1995), Artificial neural network modeling of the rainfall-runoff process, Water Resour. Res., 31(10), 2517 – 2530.

Jothityangkoon, C., M. Sivapalan, and D. Farmer (2001), Process controls of water balance variability in a large semi-arid catchment: Downward approach to hydrological model development, J. Hydrol., 254, 174 – 198.

Martinez, G.F., Gupta, H.V., 2010. Toward improved identification of hydrological models: A diagnostic evaluation of the "abcd" monthly water balance model for the conterminous United States. Water Resour. Res., 46(8).

Sivapalan, M., G. Blöschl, L. Zhang, and R. Vertessy (2003), Downward approach to hydrological prediction, Hydrol. Processes, 17, 2101–2111.

Young, P. C., and S. Parkinson (2002), Simplicity out of complexity, in Environmental Foresight and Models: A Manifesto, edited by M. B. Beck, Elsevier Science, The Netherlands, 251–294.

**Referee #2**

This paper presents a hydrologic emulator (HE), built upon the pre-existing "abcd" model, designed for use in global modeling applications such as IAMs. Both a distributed, gridded version and a lumped, water basin scale version are described and evaluated. The HE is tested against a baseline model of climatological monthly mean runoff. The HE is calibrated and validated against the VIC model, and its computational efficiency is assessed. The development of a computationally efficient, open-source global hydrologic model emulator is timely and useful to the modeling community, as many inter-disciplinary multi-modeling studies are utilizing global hydrologic models. While this paper is well-written and will add a valuable

model to the hydrology literature, there are some improvements that should be made before publication. These are described below, in addition to some suggestions.

Criticism related to previous reviewer's comments:

The current manuscript has successfully addressed most of the concerns of the previous reviews, but some improvements are still needed, and some concerns still need to be addressed.
1. The comparison of VIC runoff to GRDC data (Fig. S1) addresses the concern raised by a previous reviewer that VIC may not be an accurate model of global runoff.
However, there needs to be a few improvements to this assessment:
1) The acronyms GRDC and UNH/GRDC need their full names spelled out, and the GRDC needs to be properly cited.
**Response 3**: We have spelled out the full names for the first use, and have added citation for GRDC in the main text (lines 223-227):

*"The VIC runoff product compares well to other products (see Fig. S1, S2), including the University of New Hampshire/Global Runoff Data Centre (UNH/GRDC) runoff product (Fekete and Vorosmarty, 2011; Fekete et al., 2002) …… The scatterplot pattern of the VIC long-term annual runoff product vs. the GRDC product (GRDC, 2017) matches well with that of the UNH/GRDC runoff vs. the GRDC product…"*

References:

Fekete, B., Vorosmarty, C., 2011. ISLSCP II UNH/GRDC Composite Monthly Runoff. ISLSCP Initiative II Collection, edited by: Hall, FG, Collatz, G., Meeson, B., Los, S., Brown de Colstoun, E., and Landis, D., Data set, available at: http://daac.ornl.gov/, from Oak Ridge National Laboratory Distributed Active Archive Center, Oak Ridge, Tennessee, USA, doi, 10.

Fekete, B.M., Vörösmarty, C.J., Grabs, W., 2002. High‐resolution fields of global runoff combining observed river discharge and simulated water balances. Global Biogeochem. Cycles, 16(3).

GRDC, BfG The GRDC - Global Runoff Database. Available at: http://www.bafg.de/GRDC/EN/01_GRDC/13_dtbse/database_node.html. Accessed 09/13/2017

2) The top three panels of Fig. S1 look as though they are comparing UNH-GRDC (y-axis) to GRDC (x-axis). Such a comparison is not needed, and irrelevant, as the model UNH-GRDC product is calibrated to the GRDC data. Likely the figure is supposed to show VIC runoff vs GRDC runoff. Either the axis labels must be corrected, or the comparison needs to be redone.
**Response 4**: The upper panel of Figure S1 compares UNH-GRDC runoff product to GRDC data, and the lower panel compares VIC runoff to GRDC data. The point here is: the scatterplot patterns of the upper panel matches well with the counterparts of the lower panel, which means the behavior of the VIC runoff product is similar to that of the UNH/GRDC product, suggesting the reasonableness of the VIC runoff product, because the UNH/GRDC runoff is calibrated with the GRDC observations. This point has been clarified in the main text (lines 226-232):

*"The scatterplot pattern of the VIC long-term annual runoff product vs. the GRDC product (GRDC, 2017) matches well with that of the UNH/GRDC runoff vs. the GRDC product (streamflow is transferred to the same unit as runoff via dividing by the basin area), which means the behavior of the VIC runoff product is similar to that of the UNH/GRDC product. Further, the correlation coefficient of the VIC and the UNH/GRDC long-term annual runoff is as high as 0.83 across the global 235 basins (Fig. S2). This suggests the reasonableness of VIC runoff product, because the UNH/GRDC runoff is calibrated with the GRDC observations."*

References:

GRDC, BfG The GRDC - Global Runoff Database. Available at: http://www.bafg.de/GRDC/EN/01_GRDC/13_dtbse/database_node.html. Accessed 09/13/2017

 3) Why are these three basins chosen? The authors do not provide sufficient evidence that these basins are representative of global runoff patterns. The authors should either make this argument, or provide analysis of more basins. A map showing the r2 values of monthly runoff in VIC vs GRDC or UNH-GRDC would be most informative, as it would show regions in which VIC is most (and least) accurate.
**Response 5**: The three basins are located in three different climate zones: tropical, temperate and Arctic, which provides a glimpse of performance of the VIC runoff products at different climate zones.  Further, the scatter plot of VIC runoff product vs. UNH/GRDC runoff across global 235 basins in Figure S2 clearly indicate a strong correlation (correlation coefficient r=0.83) between these two products, which further corroborate the reasonableness of the VIC runoff product. More importantly, the VIC used in this study is merely an example to illustrate the capability of the HE in emulating global hydrological models (GHMs), and its use is not bundled with the VIC and can be used to mimic other GHMs of interest. Although we provide some assessment for the credibility of the VIC runoff product, examining performance of the VIC model is outside the focus of this study.

4) Why is the comparison only made for the period 1986-1995? GRDC data now has observations through the year 2016.
**Response 6**: We compare the VIC global runoff product with that of the UNH/GRDC runoff product, which is only available from 1986-1995. Although GRDC data has observations till 2016, it does not have a grid-level global coverage.

5) Suggestion: the authors could include a brief discussion of the limitations of the VIC model. This is not necessary, but could be helpful to readers.

**Response 7**: We have added discussions on the VIC model as follows (lines 241-250):

*"Uncertainties arising from the runoff process in the VIC model should be acknowledged. Implementation of different runoff generation schemes (e.g. TOPMODEL) within the same modeling framework is an alternative that can be adopted in the future to explore the uncertainty range. A recent inter-model comparison study shows that the VIC model falls within the range of large model ensembles (Hattermann et al. 2017). Notably, groundwater and its interaction with river and land surface are not represented in the model. Thus, the model may not be able to fully*

*capture the hydrologic responses in areas where lateral flow and the three way streamflow-aquifer-land interactions are important. Further, vegetation dynamics and water management that may affect runoff are not considered in the model simulations. Nonetheless, the use of the HE documented here is not tied to the VIC, and it could be used to emulate other GHMs of interest.*"

References:

Hattermann, F. et al., 2017. Cross‐scale intercomparison of climate change impacts simulated by regional and global hydrological models in eleven large river basins. Clim. Change: 1-16.

2. The authors should assess the computational efficiency of the calibration processes. This would inform other users of the HE how difficult it is to re-calibrate the model to other GHMs. While not necessary for publication, it would improve the paper to re-calibrate the HE to another GHM, demonstrating the HE's flexibility and broad applicability.

**Response 8**: We have conducted an experiment for the Amazon basin to assess the computational efficiency (see Table S1) of the HE, and it provides a glimpse of the computational efficiency of the calibration processes. The related results are presented in the main text (lines 385-387):

"*Take the Amazon basin that covers a total number of 2002 0.5-degree grid cells as an example, it takes 11.05 minutes for model calibration via the GA method in the distributed scheme but only 0.16 minute for the lumped one.*"

3. There is no analysis of daily runoff simulations. Even if the model is not intended to be used for daily simulations, this should be explained explicitly in the text. Line 342 states that distributed models such as the distributed HE presented here are better than lumped models for flood peak prediction. However, flood peak prediction is only accurate at daily time steps, so this statement should either be removed, or the daily accuracy of the distributed HE assessed.

**Response 9**: We have removed the statement of better performance of the distributed models for predicting flood peak to avoid confusions. Also, we have explicitly described the "*abcd*" model in the Section 2.1.2 and stated that it uses a monthly time step (line 120):

"*The monthly "abcd" model was first introduced by Thomas (1981) to …*"

Reference:
Thomas, H., 1981. Improved methods for national water assessment. Report WR15249270, US Water Resource Council, Washington, DC.

4. For context, the authors could add a brief description of the type of work that IAMs coupled with (or including) GHMs have been used for.

**Response 10**: We have added a brief description as follows (lines 427-431):

*"For example, a follow-up work is coupling the distributed scheme of the HE with a widely-used IAM, the Global Change Assessment Model (GCAM, Edmonds et al., 1997), and then using the coupled model to investigate the impacts of a variety of land use policies on global water scarcity, where the HE is used to estimate grid-level runoff globally under different land use policies."*

Reference:

Edmonds, J., M. Wise, H. Pitcher, R. Richels, T. Wigley, and C. MacCracken. (1997) "An Integrated Assessment of Climate Change and the Accelerated Introduction of Advanced Energy Technologies", Mitigation and Adaptation Strategies for Global Change, 1, pp. 311-339

Major criticisms: must be addressed before publication

1. This model is intended to be fully open-source and user-friendly. To accomplish this goal, the authors should **include in the source package a user manual**. A good example of such a model user manual is the open source CaMa-Flood manual, available here: http://hydro.iis.utokyo.ac.jp/~yamadai/cama-flood/Manual_CaMa-Flood_v362.pdf

**Response 11**: Following the reviewer's suggestion, we have included a user's manual on the Github (https://github.com/JGCRI/hydro-emulator/blob/master/docs/he_user_manual.pdf).

2. Line 107: Where does the baseline model's climatology runoff come from? Is this based on data, or a model simulation? It needs to be described and cited.
**Response 12**: We have clarified the climatology runoff as follows (lines 109-117):

*"In this study, the baseline model is based on monthly climatology runoff, which comes from a model simulation product – i.e., the runoff product from the Variable Infiltration Capacity (VIC) model (Leng et al. 2015). Specifically, we first calculate grid-level inter-annual mean value for each of the 365 calendar days from daily runoff of the benchmark product during 1971-2010 (see Section 2.3.2), and then aggregate daily climatology runoff to monthly climatology runoff at grid-level. The baseline model here uses monthly climatology runoff for prediction. For example, if the climatology runoff for July in one grid cell is 100 mm mon$^{-1}$, then the prediction of total runoff for July of every year in that specific grid cell is 100 mm mon$^{-1}$."*

Reference:

Leng, G., Tang, Q., Rayburg, S., 2015. Climate change impacts on meteorological, agricultural and hydrological droughts in China. Global Planet. Change, 126: 23-34.

3. Lines 226 – 229: Provide data or a citation to back up the claim that discrepancies between VIC runoff and observed streamflow products are due to human activities.

**Response 13**:

Typically, the VIC model simulates runoff at natural conditions, and then a stand-alone routing model can be used to route these flows downstream, and the routing model may account for human activities such as water extractions, and reservoir operations. However, here we use the VIC runoff product under natural conditions rather than the streamflow product from the routing model as the benchmark. Further, to attribute the bias of VIC runoff to human activities is non-trivial, and would typically require paired simulations to examine whether model bias under natural conditions is reduced after consideration of human impacts, which is obviously not within the scope of this study. A recent study comparing VIC runoff under natural condition and human interventions showed that impact of human activity is comparable to that by climate change in certain regions (Haddeland et al. 2014). Thus, bias in VIC runoff may be partly attributed to the neglect of human activity in the model simulations. We have clarified it and incorporated relevant references in the main text (lines 235-240):

"*This is because the VIC model simulates runoff at natural conditions, and then a stand-alone routing model can be used to route these flows downstream (Nijssen et al., 2001). The routing model may account for human activities such as water extractions, and reservoir operations (Haddeland et al., 2014). However, here we use the VIC runoff under natural conditions as the benchmark product, which explains the discrepancies between the VIC runoff and observed streamflow products.*"

References:

Haddeland, I., Heinke, J., Biemans, H., Eisner, S., Flörke, M., Hanasaki, N., Konzmann, M., Ludwig, F., Masaki, Y., Schewe, J. and Stacke, T., 2014. Global water resources affected by human interventions and climate change. *Proc. Natl. Acad. Sci.*, 111(9), 3251-3256.

Nijssen, B.N., G.M. O'Donnell, D.P. Lettenmaier and E.F. Wood, 2001: Predicting the discharge of global rivers, *J. Clim.*, 14(15), 3307-3323

4. If the VIC simulation did not include human activities, then can the HE model be used to emulate GHMs that do include human activities such as water extractions from rivers and reservoir operation?
**Response 14**: The current version of HE can only simulate runoff under natural conditions. Representation of human activities such as water withdraws will be incorporated in future versions of HE in order to emulate GHMs that include human activities.

5. Section 2.4: Please describe the runoff range over which the model is calibrated. Does it include a good representation of extreme events? How does the distribution of runoff in the calibration period compare to potential future runoff under climate change? If there is a significant difference in these distributions, the applicability of the HE to climate change studies should be discussed.
**Response 15**: The monthly runoff ranges from 0-350 mm mon$^{-1}$ in our simulation, which accommodates the most possible range of runoff across the globe. The performance of the HE largely hinges on the performance of the global hydrological model (GHM) being emulated,

although in this study we take the VIC model as an example. If the GHM being emulated has a good performance and the HE mimics the behavior of the GHM well, presumably the HE will also simulate the water budgets well, and this could be evaluated by the users when they use the HE to emulate a specific GHM. The essence of this work is to deliver an open-source and easy-to-use hydrological emulator that can be used for emulating complex GHMs of interest, and we have proved its superiority in computational efficiency and reasonableness in predictability. In terms of the application of the HE under future climate change, it is out of the scope of this work.

6. Figure S4: Only the correlation coefficient for calibration on runoff is shown. The correlation for calibration with runoff and BFI should be included, as it is discussed in the text.

**Response 16**: Actually, the correlation for calibration with both runoff and BFI in the objective function has already been presented in Figure 3 in the main text, and now we have clarified it and have cited Figure 3 in the main text as follows (lines 310-319):

"*…our analysis indicates that the incorporation of BFI into the objective function leads to a significant improvement in the partition of total runoff between direct runoff and baseflow (Fig. 3, Fig. S4), without compromising predictability for total runoff, i.e., the global mean KGE values for modeled total runoff with or without the incorporation of BFI are almost the same (0.75 vs 0.76). Specifically, for the case of involving both the total runoff and BFI in the objective function, the correlation efficiencies (r) between the long-term annual benchmark and modeled direct runoff, and between benchmark and modeled baseflow from the lumped scheme across global basins are both 0.98 (Fig. 3), which are much higher than those of 0.86 and 0.72 in the case of only involving the total runoff in the objective function (Fig. S4).*"

7. Lines 318-325, and Fig. S5: While Fig. S5 shows maps of ET, there is no quantitative assessment of ET. I suggest either a correlation analysis, or showing a difference map along with the other maps. A difference map would be very informative, showing regions of good agreement and regions of poor agreement.

**Response 17**: We have added two percentage difference maps in ET accordingly (Figure S8 in the Supplementary Materials) and have added discussions on the performance of our modeled ET as follows (lines 349-354):

"*In addition, the percentage differences between our modeled ET and the VIC ET product further confirm that the distributed scheme significantly outperforms the lumped one (Fig. S8), with much lower differences from the VIC ET product, although discrepancies still exist in some extremely cold (e.g., Greenland) or dry regions (e.g., North Africa), which is because small differences in ET will lead to large percentage difference in those regions with low ET.*"

[Figure]

[Figure]

**Figure S8**. Spatial patterns of percentage differences in long-term annual evapotranspiration (ET, mm yr$^{-1}$) during 1971-1990 between: (a) modeled ET from the lumped scheme and the VIC ET product (lump = lumped); (b) modeled ET from the distributed scheme and the VIC ET product (dist = distributed).

8. Lines 322 – 325, and Fig.4: Figure 4 shows a good match in seasonal variation of the calibration period. It is more important to show the seasonal variation in the validation period. The text claims that the seasonal variation in ET is good, but there is no quantitative evidence of this.

**Response 18**: We have added a Figure S5 for the validation period in the Supplementary Materials to present the good performance of the "*abcd*" model in capturing seasonality, and

have cited it to support the model's good performance in simulating seasonal variations as follows (lines 333-334):

"*Furthermore, both schemes display good capability in capturing the seasonal* variations *of total runoff for both the calibration and validation periods (Fig. 4, Fig. S5).* "

[Figure]

**Figure S5**. Time series of basin-specific total runoff ($Q_{total}$) from the VIC product, the lumped and distributed "*abcd*" schemes for the world's sixteen river basins with top annual flow (Dai et al. 2009) during 2001-2010 (part of the validation period 1991-2010). $KGE_l$ and $KGE_d$ stand for KGE value for the lumped and distributed scheme, respectively.

Based on the water mass balance equation, precipitation should approximate the sum of ET and runoff given the changes in basin-scale monthly soil moisture is relatively small. Thus, given the good match of seasonal variations in runoff between the VIC product and our modeled runoff, it is reasonable to infer the good match for the modeled ET, so we avoid the redundancy of presenting another figure of seasonal variations for ET. Related explanations are presented in the main text (lines 354-357):

"*Given the changes in basin-scale monthly soil moisture is relatively small, precipitation should approximate the sum of ET and runoff according to the water mass balance, the good predictability of seasonality in runoff as illustrated in Fig. 4 also reflects similar performance for ET.*"

Minor criticisms: suggestions that are not essential for publication

1. Figure 4: The color scheme is good, as blue and black are similar, and the light green is hard to see. Choosing different colors, or even using some dashed lines or other symbols would improve this figure.

**Response 19**: We have changed the colors of the lines in Figure 4 to make it more discernible.

2. Figure 5: Showing a difference map, especially between VIC and the distributed model, would be very informative.

**Response 20**: We have added two percentage difference maps between the modeled runoff and the VIC runoff product (Figure S6 in the Supplementary Materials), and also have added discussions in the main text (lines 337-342):

"*Likewise, overall much lower percentage differences between the modeled runoff from the distributed scheme and the VIC runoff product than those between the VIC and the lumped one further corroborate the significantly better performance of the distributed scheme (Fig. S6). Both schemes still show large percentage differences in some dry (e.g., North Africa) or cold regions (e.g., Tibet Plateau). This is because the runoff there is at a low magnitude and thus small changes in runoff will lead to large percentage differences.*"

[Figure]

**Figure S6**. Spatial patterns of percentage differences in long-term annual total runoff (mm yr-1) during 1971-1990 between: a) modeled runoff from the lumped scheme and the VIC runoff product (lump = lumped); b) modeled runoff from the distributed scheme and the VIC runoff product (dist = distributed).

3. While the citation for the PET calculation is given in the text, it would be useful to either cite this again within Appendix A, and/or provide the full equation for PET within Appendix A.
**Response 21**: We have added a citation for the PET calculation in the Appendix A (line 571):

"*…where PET is calculated by using the Hargreaves-Samani method (Hargreaves and Samani, 1982).*"

4. Line 257: The objective function equation needs an equation number.
**Response 22**: We have added the equation number (6) in the main text.

5. While the paper is mostly well-written, the authors should have a copy editor review the paper for detailed grammatical issues, as there are several sprinkled throughout the text. In a few places, these grammatical issues hinder the clarity of the text and should be revised. These places are:
a. Lines 164 – 166, sentence beginning with "For the baseline model…"
b. Lines 211 – 215, sentence beginning with "Second, since we have not…"

**Response 23**: We have asked a colleague who is a native English speaker to proofread the paper and have carefully addressed the grammatical issues. We have revised the two above sentences as follows (lines 167-169, 214-219):

"*For the baseline model, as documented in Section 2.1.1, every 0.5-degree grid cell of each basin has its own monthly climatology runoff estimates for each of the 12 calendar months.*"

"*Second, the simulated monthly runoff by the "abcd" model is more representative of "natural conditions" because human activities (e.g., reservoir regulations and upstream water withdrawals) are currently not represented in the model. Thus it tends to be more reasonable to compare the simulated runoff against the VIC natural runoff product rather than comparing against observed streamflow data from stream gauges (Dai et al., 2009; Wilkinson et al., 2014).*"

Reference:

Dai, A., Qian, T., Trenberth, K.E., Milliman, J.D., 2009. Changes in continental freshwater discharge from 1948 to 2004. J. Clim., 22(10): 2773-2792.

Wilkinson, K., von Zabern, M., Scherzer, J., 2014. Global Freshwater Fluxes into the World Oceans, Tech. Report prepared for the GRDC. Koblenz, Federal Institute of Hydrology (BfG),(GRDC Report No. 44. doi: 10.5675/GRDC_Report_44, 23pp.[Available from h ttp://www. bafg. de/GRDC/EN/02_srvcs/24_rprtsrs/report_44. pdf].

6. Lastly, the open source code is written in Matlab, a proprietary and costly computing software package. While most large U.S. and European universities have Matlab licenses, this platform may be cost prohibitive to some researchers, limiting the global usability of the open source model. While this is not required for publication, I would highly recommend that the authors translate this model into a fully open-source coding language such as R, Python, or C.

**Response 24**: The codes of the HE written in Matlab are available on the open-source software site Github (https://github.com/JGCRI/hydro-emulator/). In addition, the HE documented here has been translated into Python and is also freely available online. We have added clarification in the section of Code and/or data availability ():

"*In addition, the HE documented here has been translated into Python and is being incorporated into Xanthos (Li et al., 2017), which is an open-source global hydrologic model that allows users to run different combinations of evapotranspiration, runoff, and routing models. The HE will be the default runoff model used in Xanthos 2.0 and will be available on GitHub (https://github.com/JGCRI/xanthos).*"

Reference:

Li, X., Vernon, C.R., Hejazi, M.I., Link, R.P., Feng, L., Liu, Y., Rauchenstein, L.T., 2017, Xanthos – A Global Hydrologic Model, Journal of Open Research Software, 5(1), p.21.

**Referee #3**

Accepted as is

**Response 25**: We thank the referee for favorable consideration of our work.

---

## Author Response (AR3)

Response Letter

Title:        A Hydrological Emulator for Global Applications – HE v1.0.0

Journal:    Geoscientific Model Development

*We would like to thank the Editor for favorable consideration of our work. The responses to the Editor's comments are shown in* blue *font below. All the line numbers indicated refer to the main text of the revised manuscript (clean version without tracking changes).*

**Editor's comments:**
* * *
**Topical Editor Decision**: Publish subject to technical corrections (19 Feb 2018) by Bethanna Jackson

Comments to the Author:

Thanks for the careful revisions; I'm satisfied the (already positive) suggestions from the reviewers have been carefully addressed. As a personal suggestion for further work, I would be very interested to see the approach applied to further models with more spatial and temporal complexity if this is possible!

**Response 1**: We thank the Editor for accepting our manuscript with corrections. We have added some discussions for future work accordingly (lines 469-474):

"This HE could be used to emulate a wide range of models with different spatial and temporal complexities, and its performance may vary from model to model. Thus, examining and comparing the extent to which the HE could mimic the behaviors of different GHMs and LSMs is of our future research interest. In addition, future research can extend this work by systematically investigating the role of different levels of inputs and parameters on model performance in different basins across the globe."